# Recent Molecular Aspects and Integrated Omics Strategies for Understanding the Abiotic Stress Tolerance of Rice

**DOI:** 10.3390/plants12102019

**Published:** 2023-05-18

**Authors:** Babar Usman, Behnam Derakhshani, Ki-Hong Jung

**Affiliations:** 1Graduate School of Green Green-Bio Science and Crop Biotech Institute, Kyung Hee University, Yongin 17104, Republic of Korea; babarusman119@gmail.com (B.U.);; 2Research Center for Plant Plasticity, Kyung Hee University, Yongin 17104, Republic of Korea

**Keywords:** rice, abiotic stress, candidate genes, omics, breeding

## Abstract

Rice is an important staple food crop for over half of the world’s population. However, abiotic stresses seriously threaten rice yield improvement and sustainable production. Breeding and planting rice varieties with high environmental stress tolerance are the most cost-effective, safe, healthy, and environmentally friendly strategies. In-depth research on the molecular mechanism of rice plants in response to different stresses can provide an important theoretical basis for breeding rice varieties with higher stress resistance. This review presents the molecular mechanisms and the effects of various abiotic stresses on rice growth and development and explains the signal perception mode and transduction pathways. Meanwhile, the regulatory mechanisms of critical transcription factors in regulating gene expression and important downstream factors in coordinating stress tolerance are outlined. Finally, the utilization of omics approaches to retrieve hub genes and an outlook on future research are prospected, focusing on the regulatory mechanisms of multi-signaling network modules and sustainable rice production.

## 1. Introduction

Rice (*Oryza sativa* L.) is a major food crop and staple food source for half of the world’s population [1]. According to the report published by the Department of Economic and Social Affairs of the United Nations (https://www.un.org/en/desa, accessed on 3 March 2023), the world population is projected to reach almost 10 billion in 2050 and 11.2 billion in 2100. According to the Global Agricultural Productivity Index 2022 (http://www.globalagriculturalproductivity.org/, accessed on 3 March 2023), global agricultural productivity has fallen below the level needed for the sustainable growth of agricultural output. The average annual Total Factor Productivity growth rate declined from 1.99% in 2001–2010 to 1.12% in 2011–2020. As per the world’s population growth rate, rice production will need to increase by 160 million tons from the current yield (104 million tons) [2,3]. In various aspects, rice significantly promotes social stability, economic development, and food security. By providing a staple food source, rice can help stabilize communities and societies that depend on it for nutrition. Secondly, the cultivation and production of rice can provide employment opportunities for people within the agricultural sector, thereby promoting economic development and reducing poverty. For 70% of the low-income people in Asia, rice is not just a food crop but a way of life, as it is often their primary source of income. We will need about twice as much total food production in the predicted scenario to feed our fast-growing population. In past decades, breakthroughs in rice hybridization and dwarf breeding substantially enhanced grain yield and provided a roadmap for future breeding programs [4]. Climate change, including frequent extreme temperatures, rising carbon dioxide and ozone levels, and uneven rainfall, aggravates the degree of drought or salinization of agricultural lands and changes the growing environment of crops. Despite the significant increase in rice yield, plant breeders face challenges amid the fast-growing world population and the effects of climate change. The acceleration of global climate change has led to an increased incidence of abiotic stress factors, leading to a loss of yields in approximately 50% of rice-growing land [5,6]. In Asia alone, around 42 million hectares of rice-growing areas are affected by drought stress, while globally, 45 million hectares of irrigated land and 32 million hectares of rain-fed dryland are impaired by salinity stress [6]. Additionally, flooding affects over 35% of global rice-growing land [7], and rice productivity is notably impacted in rain-fed areas (which represent more than 50% of total rice-growing land) due to the prevalence of floods, water deficits, and salt stresses [8,9]. As per estimation, countries (China, India, the United States, and Indonesia) where more than 70% of the global population resides may experience increased flood risk due to changing seawater levels and extreme weather events [10]. For instance, a four-degree increase in temperature could lead to a decline in crop yields of 15–35% in Africa and Asia, with estimates of a 25–35% yield reduction in the Middle East [11]. It is reported that elevated salinity levels affect 7% of the world’s land area, leading to an annual economic loss of USD 27.2 billion due to adverse environmental impacts [12]. In addition, it is projected that salinity stress will impact over 30% of cultivable lands globally by 2050 [13], with the potential for further increases due to changes in seawater levels. Therefore, cloning important genes related to abiotic stress tolerance, analyzing the molecular mechanisms of stress responses, and further mining their breeding application value hold important theoretical and practical significance for rice breeding and food security.

The fine-mapping and cloning of several genes that are responsible for tolerance to various abiotic stresses has laid the foundation for yield improvement and enriched the genetic resources of rice [14,15]. Additionally, integrative omics approaches have efficiently determined the genetic variability of the available germplasm resources and provided the basis for rice breeding. The regulatory genes that are responsible for important agronomic traits have been reported through the efficient utilization of genome-wide association studies (GWAS) and integrative omics approaches, which is significant in rice breeding [16,17]. Mainly, the massive utilization of multi-omics methods has provided innovative insights into several biological topics and has been utilized for the genetic improvements to rice [18,19]. Their use is particularly vital for reducing agricultural costs and providing environmental protection.

In short, addressing food security is an endless and challenging task. Here, we review the key achievements made in rice genetics and breeding for environmental stress tolerance in past years, which will help provide systematic and comprehensive information and theoretical support for developing rice with improved yield and tolerance to multiple stresses.

## 2. Recognition and Signaling of Abiotic Stress

Abiotic stresses severely harm agricultural production and cause extreme deterioration of the ecological environment. When plants are stressed, molecular and morphological changes occur, directly or indirectly affecting plant growth and development. Abiotic stress decreases the final yield by severely affecting the physiology and morphology of crops at the vegetative and reproductive stages. Adversities, such as extreme temperatures, drought, and salinity, are often interconnected. In the past decade, abiotic stresses have occurred more commonly because of global climate change [20,21]. These stresses hamper water uptake and nutrient absorption and disturb plant growth and development, significantly decreasing the germination percentage, growth rate, leaf size, and productive tillers [22]. At the reproductive stage, rice displays more susceptibility to extreme temperatures, and both low and high temperatures lead to spikelet sterility and decreased grain filling [23,24]. Since plants are sessile organisms, they have evolved complex mechanisms to combat stresses by activating cellular machineries, stress avoidance, and recovery from stress [25]. After a stress attack, plants perceive the signals, transduce them to various cellular compartments, and establish a response mechanism to alleviate the effects of episodic stress events.

Plant responses to abiotic stress generally begin with signal perception. After sensing the abiotic stress, the plasma membrane produces secondary signaling molecules, such as reactive oxygen species (ROS) and inositol phospholipids. The secondary messenger stimulates the intracellular membrane by regulating the intracellular Ca^2+^ level, which then initiates a protein phosphorylation cascade to generate phosphorylated proteins that are directly participating in cytoprotection or regulating the transcription of specific stress-responsive target genes [26]. Some gene products are involved in the production of regulatory factors, such as abscisic acid (ABA) and ethylene, which activate the expression of transcription factors (TFs) [27]. TFs bind to target gene-promoter sequences, further activating or inhibiting downstream functional gene expression, and ultimately play a regulatory role in abiotic stress responses [28]. Revealing abiotic stress response mechanisms and cultivating new varieties of stress-resistant crops will help achieve sustainable agricultural development and ensure food security for the growing world population. Here, we discuss the transcriptional networks that mediate the rice plant’s response to different abiotic stresses, with a particular emphasis on the candidate genes and TFs that are involved in abiotic stress responses and play a critical role in conferring stress tolerance. A list of important candidate genes with their genomic locations is given in Table 1, and their functions are discussed in subsequent sections.

### 2.1. Low Temperature

In rice, a low temperature below 15 °C directly affects the germination percentage, seedling vigor, tillering, reproduction, and grain maturity [29]. Moreover, many physiological, biochemical, and molecular changes occur during cold acclimation, including the activation of antioxidant systems, the synthesis and accumulation of cryoprotectants, and mechanisms protecting and stabilizing cell membranes [30]. To maintain cell membrane stability, the level of unsaturated phospholipids in the membrane composition increases, and cells accumulate sucrose- and proline-rich osmotic molecules and antifreeze proteins, which trap water molecules by creating hydrogen bonds. Cold stress changes the cell membrane’s fluidity and affects the structure and activity of membrane-localized proteins, triggering the Ca^2+^ influx, which is essential for inducing the expression of temperature-responsive genes [31]. Studies have confirmed that the rice cold sensor (*COLD1*) binds to the *rice G protein α subunit 1* (*RGA1*) to mediate cold sensing and low-temperature-induced extracellular Ca^2+^ influx in rice [14] (Figure 1A). In addition to *COLD1*, the rice CBL-interacting protein kinase 7 (*OsCIPK7*) is thought to sense low-temperature signals by regulating the conformation of its kinase domain and Ca^2+^ influx [32]. However, the mechanism by which *COLD1* and *OsCIPK7* regulate Ca^2+^ influx under low-temperature stress remains unclear. Over the past decades, many chilling stress-related genes have been identified in rice using different genetic approaches; however, few have been cloned and characterized [33].

Chlorophyll content and fluorescence are important indicators for evaluating plant stress tolerance [34]. In rice, low temperatures reduce chlorophyll synthesis and chloroplast formation, confirming that chlorophyll-related changes are essential indicators of rice tolerance to low temperatures [35]. Studies have demonstrated that overexpressing *OsiSAP8* in rice can significantly improve the plant chlorophyll content and cold tolerance [36]. In rice, the cell membrane is the first to perceive low temperatures and chilling damage, and its physicochemical properties are sensitive to low temperatures, resulting in intracellular electrolyte leakage. Therefore, the electrolyte leakage rate is often an important indicator of whether plants can tolerate low temperatures [37]. It was revealed that the overexpression of *OsNAC5* results in decreased electrolyte leakage, thus indicating tolerance to low temperatures [38].

Under cold stress in rice, the intracellular oxygen metabolism is imbalanced, and ROS are generated, triggering membrane lipid peroxidation and resulting in cell membrane system damage [39]. ROS signaling can activate stress-responsive genes and downstream signaling pathways that help plants cope with cold stress, such as antioxidant defense mechanisms, osmolyte accumulation, and membrane remodeling. ROS also promote polyunsaturated fatty acid degradation and malondialdehyde (MDA) production, further damaging plant tissues and cells [40]. The protective mechanism of rice against oxidative stress has two major systems: the enzymatic and the non-enzymatic systems. The enzymatic system includes various antioxidant enzymes catalyzing ROS-scavenging reactions. Among them, superoxide dismutase (SOD) and catalase (CAT) are the two most effective antioxidant enzymes, converting superoxide anion and hydrogen peroxide (H_2_O_2_) into water and oxygen molecules, thus reducing ROS damage to cells [41]. Non-enzymatic systems also include various antioxidants, among which reduced glutathione (GSH) and ascorbic acid are the most essential [42,43]. In rice, overexpressing the ascorbate peroxidase gene *OsAPXa* can increase ascorbate peroxidase activity at low temperatures and reduce lipid peroxidation and MDA levels, improving the seed setting rate of rice at low temperatures [44].

Under cold stress, rice accumulates large amounts of soluble sugars, such as sucrose, hexose, raffinose, glucose, fructose, and trehalose. Soluble sugars can act as osmotic regulators of cells under low temperatures, stabilizing cell membranes and protoplasmic colloids. In addition, sugars also provide a carbon skeleton and energy for synthesizing other organic substances [45,46]. Overexpressing the trehalose synthesis genes *OsTPP1*, *OsTPP2*, and *OsTPS1* can significantly improve rice tolerance to low temperatures [47].

In rice, when the pollen mother cell meiosis encounters low temperatures, sugars—including sucrose, glucose, and fructose—accumulate in the anthers, and simultaneously, the activity of sucrose-degrading enzymes decreases, and the expression of monosaccharide transporters is downregulated. This results in an insufficient supply of sucrose to the tapetum and pollen grains, causing pollen sterility [48]. A previous study observed that externally applying sucrose could significantly improve the fertility of rice pollen under low temperatures and increase the seed setting rate [49]. Rice also accumulates a large amount of proline at low temperatures. Proline is widely involved in osmotic regulation and carbon and nitrogen metabolism and protects most enzymes from denaturation and inactivation [50]. At the same time, proline also stabilizes polyribosomes and maintains protein synthesis [51]. Under adverse conditions, proline can also remove the excess hydrogen ions (H^+^) produced by the stress reaction and maintain the optimal pH for aerobic respiration in the cytoplasm [52]. In addition, proline binds to proteins through its hydrophobic group to improve the hydrophilicity of proteins [53]. In rice, the overexpression of *OsCOIN* can lead to a significantly increased proline content and enhanced low-temperature tolerance [54].

ABA is essential in low-temperature adversity [55]. Under cold stress, maintaining a relatively low level of ABA is beneficial for improving the stress tolerance of rice. Overexpressing the rice ABA metabolism gene *OsABA8ox1* reduced ABA levels in rice seedlings and improved the tolerance to low temperatures [56]. The ABA signaling pathway comprises the ABA receptors PYR/PYL/RCAR (pyrabactin resistance/pyrabactin resistance-like/regulatory component of the abscisic acid receptor), negative regulator type 2C protein phosphatase (PP2C), positive regulator four core components, including SNF1-related protein kinase 2 (SNF1-related protein kinase 2, SnRK2), and the TFs AREB/ABF that form a dual negative regulatory system [57,58,59,60].

Under low-temperature conditions, endogenous ABA increases and binds to PYR/PYL/RCAR. Consequently, PYR/PYL/RCAR interacts with PP2C, thus inhibiting its binding to SnRK2. SnRK2 can phosphorylate TFs and activate ABA-responsive gene expressions, improving plant tolerance to low temperatures. However, under normal conditions, the endogenous ABA content is unchanged, and the interaction between PP2C and SnRK2 prevents the latter from phosphorylating downstream substrates, repressing the expression of ABA-responsive genes [61]. The ABA receptor *OsPYL9* positively regulates ABA signaling, and its overexpression can significantly improve cold tolerance in rice [62]. In addition to the core component PYL-PP2C-SnRK2-ABF, the components of the ABA signaling pathway include Ca^2+^, ROS, nitric oxide (NO), phospholipid molecules, and other kinases, such as mitogen-activated protein kinases (MAPK) [26].

The C-repeat-binding factor/dehydration-responsive element-binding factor (CBF/DREB1) is an essential TF part of ABA-independent low-temperature response signaling pathways. CBF belongs to a subfamily of the AP2/ERF (APETALA 2/ethylene responsive) TF family. The AP2/ERF family is divided into four subfamilies: AP2, ERF, DREB, and RAV (related to VP1/ABI3) [63]. CBF contains a conserved AP2 domain, which can bind to the promoter region of low-temperature-responsive genes (CORs) containing the core element CCGAC (also known as CRT, C-repeat) under low temperatures and activate the transcription of CORs. The *CBF* gene is usually regulated by the bHLH-like TF (ICE1). Therefore, the low-temperature regulation pathway is also called the ICE-CBF-COR pathway [64]. Previous studies have demonstrated that overexpressing *OsDREB1D* and *OsDREB1F* of the DREB1 subfamily in rice can improve the tolerance of rice to low temperatures [65].

The mitogen-activated protein kinase *OsMAPK3* phosphorylates OsbHLH002/OsICE1, thus reducing its ubiquitination level and promoting the accumulation of active OsbHLH002. Later on, the expression of the downstream gene *OsTPP1* is activated by OsbHLH002, which finally increases the content of trehalose and improves rice tolerance to low temperatures [66]. In addition, some *CBF* genes are induced by ABA, such as *OsDREB1F*. *OsDREB1F* is involved in both ABA-independent and -dependent signaling pathways [67], suggesting that the ABA-dependent signaling pathway overlaps with the ICE-CBF-COR pathway [68]. Under cold stress, at the booting stage, the endogenous gibberellins (GA) content in rice decreases. A previous study demonstrated that mutants of *sd1* and *d35* were sensitive to low temperatures and that external GA treatment could improve the pollen fertility of mutants at low temperatures [49]. The recently cloned *HAN1* is crucial for rice cold tolerance at the seedling stage. *HAN1* encodes an oxidase that fine-tunes the JA-mediated chilling response by catalyzing the conversion of biologically active jasmonoyl-L-isoleucine (JA-Ile) to the inactive form 12-hydroxy-JA-Ile (12OH-JA-Ile). Functional nucleotide polymorphism in the *HAN1* promoter increases putative MYB cis-elements in the allele of temperate *japonica* rice, enhancing its cold tolerance [69,70].

Several other genes that are involved in cold stress tolerance have been cloned and functionally identified in rice. *qLTG3-1* encodes a conserved glycine-rich (GRP) domain, and the three sequence variations in the coding region determine the strength or weakness of seed germination under low temperatures in rice [71]. In a study, it was observed that a single SNP in *OsGSTZ2* was responsible for amino-acid differences and was essential for improving low-temperature tolerance in rice at the seedling stage [72]. *LTG1* encodes casein kinase I, and the amino acid substitution at position 357 (I357K) in the coding region has important effects on the growth rate, heading date, and rice yield under low temperatures [73]. *Ctb1* encodes a protein containing an F-box domain, which interacts with the E3 ubiquitin ligase subunit Skp1 and participates in low-temperature signaling in the ubiquitin–proteasome pathway [74]. *CTB4a* encodes a conserved leucine-rich repeat receptor-like kinase LRR-RLK (leucine-rich repeat receptor-like kinase), which interacts with the β subunit AtpB of ATP synthase and affects its activity. The three SNPs (−2536, −2511, and −1930 upstream of ATG) in the *CTB4a* promoter region determine the tolerance of different rice cultivars at low temperatures [33]. *OsbZIP73* positively regulates low-temperature tolerance at the seedling stage, and the interaction between *OsbZIP71* and *OsbZIP73* modulates the ROS and ABA levels in response to cold stress [75]. In rice, *NUS1* is mainly expressed in immature leaves and upregulated under cold stress, and *nus1* mutants display impaired chloroplast rRNA accumulation and repressed transcription/translation capacity [76]. Other essential genes, such as *TEMPERATURE-SENSITIVE VIRESCENT* (*TSV*), *WHITE STRIPE LEAF 5* (*WSL5*), and *OsCYP20-2*, can protect rice from chilling stress by regulating different chloroplastic and photosynthetic genes [77,78,79]. Significant progress has been made for cold stress tolerance, but our knowledge on plant responses at the single-cell level remains scarce.

**Table 1 plants-12-02019-t001:** List of key genes involved in abiotic stress tolerance of rice.

Gene Symbol	Gene Name	Locus ID	Position ^1^	Position ^2^	References
**Cold stress**					
*OsCOLD1*	*Chilling tolerance divergence 1*	*LOC_Os04g51180*	30311519–30316303	30311574–30316221	[14]
*OsCIPK7*	*CBL-interacting protein kinase 7*	*LOC_Os03g43440*	24226224–24227930	24226372–24227930	[32]
*OsiSAP8*	*Stress associated protein gene 8*	*LOC_Os06g41010*	24491979–24494238	24491993–24493907	[36]
*OsNAC5*	*NAC domain transcription factor 5*	*LOC_Os11g08210*	4299149–4301783	4299277–4301784	[38]
*OsAPXa*	*Ascorbate peroxidase 1*	*LOC_Os03g17690*	9843327–9846747	9843336–9846670	[44]
*OsTPP1*	*Trehalose-6-phosphate phosphatase 1*	*LOC_Os02g44230*	26767603–26771633	26767607–2677162	[47]
*OsCOIN*	*Cold-inducible*	*LOC_Os01g01420*	209771–214229	209771–214173	[54]
*OsPYL9*	*Pyrabactin resistance-like 9*	*LOC_Os06g36670*	21556404–21557334	21556404–21557283	[62]
*OsDREB1D*	*Dehydration responsive element-binding protein 1D*	*LOC_Os06g06970*	3310866–3311822	3310919–3311822	[65]
*OsHAN1*	*Salt overly sensitive 1*	*LOC_Os12g44360*	27494401–27508851	27495775–27508468	[69]
**Heat stress**					
*OsHSP26.7*	*Heat shock protein 26*	*LOC_Os03g14180*	7697015–7698284	7697015–7698027	[80]
*O* *sHSP17.7*	*Small heat shock protein 17.7*	*LOC_Os03g16040*	8838031–8838510	8837821–8838527	[81]
*OsTT1*	*Thermo-tolerance 1*	*LOC_Os03g26970*	15420148–15424724	15420151–15424562	[82]
*OsGIRL1*	*Gamma-ray-induced LRR-RLK1*	*LOC_Os02g12440*	6487390–6490577	6488457–6490577	[83]
*OsNSUN2*	*NOP2/SUN domain family member 2*	*LOC_Os09g29630*	18013221–18020628	18013221–18020611	[84]
*OsTCM5*	*Thermo-sensitive chlorophyll-deficient mutant 5*	*LOC_Os05g34460*	20433366–20437932	20433366–20437932	[85]
*OsEG1*	*Extra glume 1*	*LOC_Os01g67430*	39177169–39178676	39177169–39178676	[86]
*OsRBG1*	*Rice big grain 1*	*LOC_Os11g30430*	17694857–17696042	17694769–17696076	[87]
*OsANN1*	*Annexin 1*	*LOC_Os02g51750*	31698161–31700503	31698204–31700438	[88]
**Drought stress**					
*OsDRO1*	*Deeper rooting 1*	*LOC_Os09g26840*	16307780–16310837	16307780–16310837	[89]
*OsASR5*	*Abscisic acid stress and ripening 5*	*LOC_Os11g06720*	3278435–3279425	3278451–3279419	[90]
*OsDST*	*Drought and salt tolerance*	*LOC_Os03g57240*	32645456–32647051	32645695–32646908	[91]
*OsJAZ1*	*Jasmonate ZIM-domain protein 1*	*LOC_Os04g55920*	33306461–33310232	33306468–33310169	[92]
*OsEPF1*	*Epidermal patterning factor 1*	*LOC_Os04g54490*	32414780–32415613	32414780–32415613	[93]
*OsCPK9*	*Calcium-dependent protein kinase 9*	*LOC_Os03g48270*	27467403–27472759	27467413–27472746	[94]
*OsITPK2*	*3,4-trisphosphate 5/6-kinase 2*	*LOC_Os03g12840*	6901924–6907409	6902118–6907409	[95]
**Salt stress**					
*OsSAPK4*	*Stress/ABA-activated protein kinase 4*	*LOC_Os01g64970*	37710241–37715296	37710241–37714835	[96]
*OsMAPK44*	*Mitogen-activated protein kinase 44*	*LOC_Os05g49140*	28188762–28194025	28188894–28194022	[97]
*OsLOL5*	*LSD1-like-5*	*LOC_Os01g42710*	24292537–24299697	24294290–24299500	[98]
*OsBADH1*	*Betaine aldehyde dehydrogenase 1*	*LOC_Os04g39020*	23171426–23176369	23171516–23176332	[99]
*OsKAT1*	*Shaker potassium channel 1*	*LOC_Os01g55200*	31761223–31763887	31761223–31763887	[100]
*OsHAK5*	*High-affinity potassium (K^+^) transporter 5*	*LOC_Os01g70490*	40825681–40830301	40825678–40830191	[101]
*OsVP1*	*Viviparous 1*	*LOC_Os01g68370*	39723155–39726988	39723171–39726984	[102]
**Osmotic stress**					
*OsP5CS1*	*Pyrroline-5-carboxylate synthetase 5*	*LOC_Os05g38150*	22374029–22381039	22374029–22380820	[103]
*OsPPa6*	*Inorganic pyrophosphatase 6*	*LOC_Os02g52940*	32374870–32378546	32374870–32378165	[104]
*OsCCD1*	*Carotenoid-cleavage dioxygenase 1*	*LOC_Os12g44310*	27464735–27472036	27464832–27471667	[105]
*OsANN10*	*Annexin 10*	*LOC_Os09g27990*	16999259–17001374	16999461–17001374	[106]
*OsCSLD4*	*Curled leaf and dwarf 1*	*LOC_Os12g36890*	22602880–22607315	22602934–22607307	[107]
*OsPP65*	*Protein phosphatase 65*	*LOC_Os04g37660*	22389303–22393831	22389359–22394048	[108]
*OSCA1*	*Osmolality-sensing ion channel 1*	*LOC_Os01g45274*	25692717–25705090	25696671–25705077	[109]
**Submergence stress**					
*OsSUB1B*	*Submergence 1B*	*LOC_Os09g11480*	6404474–6406039	6404482–6406039	[110]
*OsSUB1C*	*Submergence 1C*	*LOC_Os09g11460*	6387891–6389789	6387891–6389789	[110]
*OsEIL1*	*Ethylene insensitive3-like gene 1A*	*LOC_Os03g20790*	11776086–11778008	11774230–11778954	[111]
*OsACE1*	*Accelerator of internode elongation 1*	*LOC_Os03g22510*	12929937–12930797	12929937–12930797	[112]

The position ^1^ and position ^2^ correspond to Rice Genome Annotation Protect (http://rice.uga.edu/ (accessed on 4 April 2023)) and RAP-DB (https://rapdb.dna.affrc.go.jp/index.html (accessed on 4 April 2023)) genome browsers, respectively.

### 2.2. Heat Stress

High temperatures above 35 °C can negatively impact plant growth, pollen viability, fertilization ability, and grain filling [113]. At the same time, high temperatures inhibit photosynthesis and reduce water content, negatively impacting plant cell division and growth [114]. At present, numerous high-temperature-related genes have been cloned in rice. They can be roughly divided into different categories, including heat shock proteins (HSPs), heat shock TFs (HSTFs), stress-related TFs, and others, and they play a role in temperature sensing and stress responses (Figure 1B).

HSPs have a wide variety and rich content in plants and are major molecular chaperone proteins. HSPs help with the correct folding of proteins and assist their transmembrane transfer under plant stress, thus enhancing the plants’ stress tolerance. About 17 proteins of this type have been reported in rice, and most are reportedly induced by heat shock only. Their specific biological functions are not clear yet [115,116]. Among them, *OsHSP26.7* encodes a chloroplastic HSP, which protects chloroplasts from oxidative damage caused by high temperatures and ultraviolet light [80]. *OsHSP17.7* was isolated from heat-stressed rice, and HSA32 and HSP101 were proved to interact to form a positive feedback regulation loop with a post-transcriptional role [81].

HSTFs form the regulatory hubs of the heat stress response, responsible for the signal transduction and activation of heat shock protein expression. Among the ~25 *HSTFs* in rice, ~13 are induced by heat shock; however, research on their functions is still lacking, and the molecular mechanism leading to their expression remains to be revealed [117]. *OsHsfA2d* encodes two splice isomers, which function under high-temperature stress and help cells establish a protein folding balance [118]. *OsHsfA2e* is localized in the nucleus and displays C-terminal transcriptional activity. Its higher expression can significantly enhance Arabidopsis environmental stress tolerance [119]. *OsHsfA2a* has multiple transcripts that are essential for rice growth and stress responses [120].

In addition to HSTFs, other TFs containing stress elements are involved in high-temperature stress. *OsDREB1B* is an *AP2/EREBP* TF whose expression is altered by temperature variations [121]. *OsWRKY11* encodes a TF containing a WRKY domain, and under the HSP101 promoter, it can significantly improve rice heat and drought tolerance [122]. *OsAREB1* and *OsbZIP60* encode a *bZIP* TF, and their expression is affected by high temperatures. The ethylene response factor (HYR) is a key regulator of the direct activation of photosynthesis and can also regulate downstream carbon metabolism genes and affect morphology–physiology under drought and high-temperature stress, stabilizing the rice yield [123]. *SNAC3* is an NAC TF that can significantly enhance high-temperature tolerance by mediating the metabolism of reactive oxygen species (ROS) [124]. In rice, the OsMYB55 TF is induced by high temperatures, and its overexpression can significantly improve high-temperature tolerance and grain yield [125].

Additionally, several enzymes are involved in temperature stress and other biological pathways in rice. *GAD3* and *OsGSr* encode a glutamate decarboxylase and a glutamine synthase, respectively, which are highly expressed after being activated by OsMYB55 to promote the synthesis of stress-related amino acids and have an essential contribution to the high-temperature tolerance of rice [125,126]. *OsHTAS* encodes a ubiquitin ligase located in the nucleus and cytoplasm. It mediates the hydrogen peroxide-induced stomatal opening and closing and has a positive regulatory effect on heat tolerance in rice [127]. *OsTT1* was identified in African rice, which has a leucine-rich repeat receptor encoding a 26S proteasome α2 subunit. It is involved in the ubiquitination and degradation of toxic proteins and is essential to the high-temperature response of rice [82]. *OsGIRL1* encodes a leucine-rich repeat receptor protein kinase, and its overexpression confers an enhanced seedling survival rate under heat stress [128]. *TOGR1* is a DEAD-box RNA helicase involved in high-temperature responses and the processing of rRNA precursors, which is essential for cell proliferation at high temperatures [83]. *AET1* is a histidine *tRNA guanylyltransferase* regulating the translation process of ribosomal proteins and is essential for maintaining protein translation stability at high temperatures [129]. *OsNSUN2* encodes an mRNA m^5^C methyltransferase, which maintains the normal growth and development of rice under high temperatures by regulating mRNA translation efficiency [84]. *PGL* encodes chlorophyll a oxidase 1, mainly expressed in green tissues, whose mutants are sensitive to heat stress [130]. *TCM5* is a chloroplast-targeted Deg protease involved in chloroplast development under high temperatures and is essential for the functional maintenance of the photosynthetic system domain [85].

*EG1* is a high-temperature-mediated mitochondrial lipase that can trigger the expression of floral organ genes under high temperatures and maintain the stability of floral organ development [86]. *OsAPX2* is a rice cytoplasmic ascorbate peroxidase gene involved in heat-induced ascorbate peroxidation in rice [131]. OsGSK1 is a homologous protein of BIN2, a key regulator of the Arabidopsis BR signaling pathway, and its knockout mutants display sensitivity to heat stress [132]. *GSA1* encodes a rice glycosyltransferase UGT83A1 with a broad-spectrum of glycosyltransferase activity and regulates the synthesis of flavonoid metabolites, rice yield, and stress resistance [133].

The accumulation of soluble sugars such as glucose and fructose, and non-soluble sugars such as starch, is an important mechanism for plants to cope with heat stress [134]. Studies have shown that a high accumulation of sugars can inhibit photosynthesis [135], possibly due to the feedback regulation of sugar accumulation in the Calvin cycle and Rubisco activity [136]. To overcome the inhibitory effect of high sugar accumulation on photosynthesis, several strategies have been proposed. For example, it has been suggested that the application of exogenous sugar-metabolizing enzymes, such as invertases and sucrose synthases, can enhance photosynthesis under heat stress by reducing the sugar accumulation [137]. In addition, the manipulation of sugar transport and allocation pathways, such as the overexpression of sugar transporters or starch synthesis enzymes, has also been proposed to reduce sugar accumulation and maintain photosynthesis under heat stress [138]. In rice, some key genes have been identified that play important roles in sugar accumulation and photosynthesis under heat stress. For example, the expression of a sucrose transporter *OsSUT1* is upregulated under heat stress, leading to an increased sugar accumulation and decreased photosynthesis [139]. Two heat-responsive genes, *ONAC127* and *ONAC129*, which encode NAM/ATAF/CUC (NAC) domain TFs were also found to be involved in the grain-filling process under heat stress. It was revealed that *ONAC127* and *ONAC129* regulate sugar transportation and abiotic stress responses, which are crucial for proper grain filling. The target genes of *ONAC127* and *ONAC129* in developing rice seeds include the sugar transporter gene *OsSWEET4* and monosaccharide transporter gene *OsMST6* [140]. Other members of few gene families like trehalose-6-phosphate phosphatase and hexokinases might also be potential targets to study sugar metabolism and photosynthesis under heat stress [47,141]. The functional analysis of these gene may help in understanding the regulation of sugar accumulation and the Calvin cycle in rice, and their manipulation may offer potential means to overcome the negative impact of sugar accumulation on photosynthesis under heat stress.

In addition to the abovementioned gene categories, others related to high temperatures have been reported. FLO2 is a protein containing a 34-peptide repeat sequence that controls the amylose content of rice seeds. The overexpression of *FLO2* causes grain enlargement and displays significant expression differences among different rice varieties under high-temperature stress [142]. *Fie1* is a fertilization-independent endosperm-autonomous gene that regulates seed size and the nighttime high-temperature sensitivity of seeds during grain filling, which is of great significance for maintaining yield under high temperatures [143]. *RBG1*, a novel positive regulator of grain size, was reported to enhance tolerance to heat, osmosis, and salt stress in rice through auxin and cytokinin pathways [87]. *OsLEA4* and *OsLEA5* encode two late embryogenesis proteins, and their expression in *Escherichia coli* led to significant high-temperature tolerance, suggesting they may contribute to stress tolerance in crop species [144]. *ZFP177* encodes an A20/AN1-type zinc finger protein induced by heat stress, and its overexpression in tobacco can improve heat stress tolerance [145]. *OsRZFP34* encodes a ring zinc finger protein induced by heat, and its loss of function leads to a smaller stomatal diameter and reduced temperature tolerance [146]. OsANN1 is a calcium-binding protein with ATPase activity, and its knockout mutants displayed sensitivity to heat stress, resulting in severe redox imbalance [88]. OsCNGC14 and OsCNGC16 are two cyclic nucleotide-gated ion channel proteins in the plasma membrane responsible for regulating the influx of cytoplasmic calcium ions; their mutants are sensitive to temperature stress [147]. *SLG1* encodes a conserved cytoplasmic tRNA 2-thiolated protein 2 (RCTU2) involved in tRNA modification and is essential in high-temperature resistance in rice seedlings and at reproductive development stages in rice [148]. OsGR-RBP4 is a glycine-rich RNA-binding protein screened from the rice cDNA library under heat shock and constitutively induced by heat stress [149].

Transcriptional modifications are crucial for environmental temperature perception and heat stress responses [150]. Recently, four genes related to post-transcriptional regulation under heat stress have been reported in rice, including RNA helicase *TOGR1*, tRNA His *guanylyltransferase AET1*, RNA methyltransferase *OsNSUN2*, and cytoplasmic tRNA 2-thiolated protein 2 *SLG1* [83,84,129,148]. Among them, TOGR1 is a chaperone protein of the small subunit complex in the nucleolus and is involved in processing rRNA precursors at high temperatures [83]. *AET1* plays a decisive role in the maturation of the precursor tRNA His on the endoplasmic reticulum and ribosomes and in later protein translation [129]. *OsNSUN2* mediates the mRNA methylation modifications of photosynthesis-related detoxification proteins in the nucleus under high temperatures and improves the translation efficiency to ensure the normal growth of rice [84].

In higher plants, the connection between ubiquitin molecules and substrates can mediate various cellular functions through the Ub/26S proteasome system, which is also crucial for plant heat tolerance [151]. Three genes involved in this pathway have been reported in rice—*OsHCI1*, *HTAS*, and *OsTT1* [82,126,127]. Among them, *HTAS* and *OsHCI1* encode E3 ubiquitin ligases. HTAS has ubiquitin ligase activity in vitro and can interact with ascorbate peroxidase to mediate stomatal opening and closing under high temperatures. *OsHCI1* accumulates in large amounts in the nucleus at high temperatures and mediates the rapid export of ubiquitination and degradation of substrate proteins along the cytoskeleton [82,126,127]. *OsTT1*, as a 26S proteasome α2 subunit, can help degrade toxic proteins more efficiently to maintain a high-temperature response [82].

In plants, the accumulation of some metabolites significantly contributes to high-temperature tolerance. In rice, the MYB55 TF regulates the expression of downstream glutamate dehydrogenase *GAD3* and glutamine synthase *OsGS1.2* and promotes the accumulation of stress-related amino acids, such as L-glutamic acid and GABA, under high temperatures [125]. In addition, the UDP glucosyltransferase gene *GSA1* can promote flavonoid and anthocyanin synthesis under heat, drought, and salt stress and enhance the stress resistance of rice [133]. The lipase gene *EG1* is located in mitochondria and plastids and mediates the mitochondrial lipase pathway under high temperatures, regulating lipid metabolism and downstream gene expression [86].

Calcium ions are essential in plant responses to abiotic stress, acting as second messengers; however, the molecular mechanism underlying the upregulation of intracellular calcium ion concentration as a key stress signal remains to be elucidated [152]. In rice, *CNGC14* and *CNGC16* act as cyclic nucleotide-gated ion channels to regulate calcium influx from the plasma membrane under high temperatures, while *OsANN1* regulates the redox balance by increasing the activity of ROS-scavenging enzymes [88,147].

In rice, *APX2* and *SNAC3* are directly related to ROS scavenging [124,131]. The ascorbate peroxidase gene *APX2* is synthesized in large amounts in the cytoplasm at high temperatures, enhancing the ability of rice to scavenge hydrogen peroxide, protecting spikelets from lipid peroxidation, and maintaining fertility [131]. The TF *SNAC3* positively regulates the expression of ROS-scavenging genes in the nucleus and enhances high-temperature tolerance in rice [124].

Hormonal signaling is also essential under high-temperature stress, and various hormonal signals interact in rice. *OsGSK1* acts as a negative regulator of BR signaling to mediate the response to heat stress in rice [132]. Another plasma membrane receptor kinase, *OsGIRL1,* regulates heat stress by responding to ABA or SA signaling [128]. The heat stress-related TF *OsAREB1* is a positive regulator of ABA signaling. Finally, *OsRZFP34* is involved in ABA-mediated stomatal opening and closing under heat stress [146].

Several high-temperature-related genes have been reported, and the signaling pathways they are involved in have gradually become clear. However, the molecular mechanism and regulatory network of high-temperature sensing, signaling, and transduction to downstream elements remain poorly known and must be the main topic for future research.

### 2.3. Drought Stress

Drought is an important abiotic factor limiting rice growth and production. The agricultural drought threat’s frequency and range are increasing worldwide, and 50% of global rice production is severely affected by it. Breeding drought-resistant rice varieties is a practical option to fight against drought stress [153]. However, drought tolerance in rice is a complex trait controlled by multiple genes and significantly interacting with the environment, which makes the drought tolerance mechanism in rice more complicated. Drought tolerance-related genes are divided into three major categories: transcriptional control, stress signaling, and membrane transport [154,155]. These genes mainly influence the molecular, physiological, and biochemical mechanisms after plants are subjected to drought stress [156]. Many work in ABA-dependent and -independent regulatory systems [157].

The root system is responsible for absorbing and translocating nutrients and water and is essential for crop drought-escaping strategies. Drought-tolerant rice varieties have a well-developed root system. By increasing the root-shoot ratio and enhancing root penetration, the plant can maintain a higher water potential under drought conditions, efficiently absorbing water in the soil and forming a stable internal environment for the normal growth of plants [155]. Under drought stress, the root system enhances the overall drought resistance of rice by improving cuticle resistance and increasing the number of root hairs, density, and depth [158]. Many genes involved in root traits in rice are also involved in drought tolerance. A major QTL *Deeper rooting 1* (*DRO1*) controls root tip cell elongation, asymmetric growth, and gravity-based downward bending of the root tip (Figure 1C). The transformation of shallow-rooted rice varieties with *DRO1* resulted in drought tolerance by developing deep roots [89]. Other QTLs, including *DRO2* and *DRO3*, also control deep rooting under normal conditions [159]. In rice, *qRL6.1* and *qRL7* are associated with root length under hydroponic conditions [160,161]. Overexpressing *OsDREB2B*, *CYP735A*, and *OsDREB1F* improves root morphological adaptations under drought conditions [155]. More than 100 association loci were identified using a GWAS study of 529 rice accessions, providing a genetic basis for drought tolerance improvement [162]. Several genes associated with osmoregulation and late embryogenesis were also identified with a positive role under water deficit conditions [156]. Other genes, such as *EcNAC67* and *OsPYL/RCAR5,* induce a higher root and shoot mass and delay leaf rolling under drought stress [163,164].

Controlling the stomatal aperture is an efficient strategy for developing drought-tolerant plants. The opening of the stomata in response to light and under normal water conditions allows water evaporation via transpiration and the entrance of CO_2_ into leaves for photosynthesis [93,165]. Under drought stress, plants close their stomata to reduce water loss and improve the water-use efficiency and survival rate. In plant species, ABA regulates stomatal movement to reduce transpiration during drought stress [57,58,166]. In rice, PYR1 and PYL proteins function as ABA receptors. The multiple mutants of group I *OsPYL1*, *OsPYL6,* and *OsPYL12* display defects in the stomatal movement and transpiration control but promote grain productivity [70]. Overexpressing *OsPYL3*, *OsPYL5*, *OsPYL9*, or *OsPYL11* improves drought tolerance but reduces yield under normal conditions [60,62,163]. Several studies reported that H_2_O_2_ is essential in stomatal conductance under drought stress in ABA-independent and -dependent manners. For example, *ABSCISIC ACID STRESS AND RIPENING5* (*OsASR5*) improves drought tolerance through a stomatal closure pathway associated with H_2_O_2_ and ABA signaling [90]. In addition, *DROUGHT AND SALT TOLERANCE* (*DST*) and *OsSRO1* work independently of ABA to regulate stomatal closure by modulating H_2_O_2_ accumulation [91,167]. *OsJAZ1* improves plant growth under drought stress and induces ABA signaling [92]. Stomatal density, size, and index are essential for improving drought tolerance. Plants overexpressing *EPIDERMAL PATTERNING FACTOR1* (*OsEPF1*) reduced their stomatal density, index, and size, improving growth, yield, and drought tolerance [93]. Previously, it was reported that *OsDRAP1* (DREB2-like) confers drought tolerance [168]. Introducing *OsLEA3-1*, *OsNAC5*, *OsWRKY47*, *OsbZIP71*, *OsNAC10*, or *OsbZIP46* in rice improves drought tolerance [169,170,171,172,173,174]. Introducing *AtDREB1A*, *EDT1/HDG11*, *OsTPS1*, or *OsMIOX* in rice improves the water-use efficiency, antioxidant activity, photosynthesis, and accumulation of osmolytes [175,176,177,178]. The transgenic plants overexpressing *OsCPK9* display improved drought tolerance through better osmoregulation and enhanced stomatal closure [94]. Overexpressing *OsDREB2A* improves the survival rate of rice plants under severe salt and drought conditions [179]. Several CDPKs (*CDPK7* and *CIPK03/CIPK12*) are involved in signal transduction pathways in the response to drought stress [180,181]. Under drought stress, reduced inositol triphosphate levels and ROS homeostasis are observed in plants overexpressing *OsITPK2* [95]. The genes from the WRKY F family are essential in rice growth in response to drought stress [182]. Multiple TFs, such as *OsMUTE*, *OsSPCHs*, *OsICEs*, and *OsFAMA,* also control stomatal movement and development in rice [183]. However, the detailed mechanism of drought tolerance by these TFs remains unknown. Stomatal development is the key parameter to control drought stress and deserves more attention to develop drought-resistant rice varieties.

### 2.4. Salt Stress

Salt stress affects rice growth and grain quality, directly affecting the market popularity and economic value of rice. Excessive soil salinity reduces the soil’s water potential, makes it difficult for plants to absorb water, and causes physiological drought. High external salt concentrations increase cell membrane permeability and electrolyte extravasation. Plants generally reduce the cellular water potential by regionalizing intracellular salts, absorbing exogeneous inorganic ions, and synthesizing organic osmotic regulators such as soluble sugar, betaine, and proline, thus enhancing plant water absorption capacity and relieving physiological drought. Under salt stress, absorbing external inorganic ions regulates cellular osmolarity in rice. Salt-tolerant rice mainly maintains cell osmotic regulation by increasing the absorption of potassium ions and reducing that of sodium ions [184].

When rice is subjected to salt stress, the excessive accumulation of Na^+^ in cells reduces the absorption of K^+^ and Mg^2+^. The cell is likely to lack phosphorus and Ca^2+^ (Figure 2A). The antagonism of salt stress on nutrient elements disrupts the growth of the apical meristem and affects chlorophyll synthesis, causing physiological disorders and hindering normal metabolism. Salt stress relatively reduces the content of K^+^, leading to cell membrane hyperpolarization and a decrease in sugar transport and osmotic pressure and affecting cell extension, growth, and development of shoots and roots. Salt stress inhibits the absorption and utilization of Ca^2+^, resulting in the blockage of cell wall formation, the inhibition of cell division, and the decline of the membrane system [185,186].

TFs positively or negatively regulate salt tolerance, and 80 TFs are upregulated under salt stress. For example, the TFs *OsDREB2A*, *OsCOIN*, *OsbZIP71*, *OsMYB2,* and *OsbZIP23* can lead to various changes in rice, including an accumulation of antioxidants and osmoprotectants and increased Na^+^ and K^+^ transporter activity [187]. Overexpressing these salt-responsive TFs enables efficient osmotic regulation, minimal oxidative damage, and increased seedling survival rates in rice [171,188]. Among the negative regulatory TFs, *OsWRKY13* delays the growth and development of plants by inhibiting the expression of the salt-responsive genes *SNAC1* and *ERD1* [189]. Overexpressing *OsWRKY45-2* resulted in a significant decrease in the survival rate of rice plants under salt stress, which was caused by the repressed expression of genes, such as *SNAC1, DREB1B*, *NCED4,* and *Rab16D,* suggesting that *OsWRKY45-2* may be a transcriptional repressor of these genes. The regulation of *OsWRKY45-2* expression by *OsWRKY13* indicated that these two TFs might work together in the same pathway to regulate rice salt tolerance jointly. In addition, rice plants displayed obvious chlorosis and dryness after overexpressing the TFs *OsABI5* and *OsABI5* [190,191]. These studies indicate that these TFs regulate the expression of different types of genes in rice through various pathways in response to salt stress. Cis-acting elements are the binding sites of TFs, and they can regulate the accurate initiation and transcription efficiency of gene transcription by binding TFs. Many cis-acting elements have been found in rice, and their specific functions have been verified with the corresponding salt-responsive TFs. For example, the promoters of most ABA-inducible genes in rice contain the ABA-responsive element ABRE (ACGTGGC), which regulates their expression under salt stress by recruiting zinc finger TFs, such as *OsBZ8* and *OsABI5* [192]. ABRE was initially discovered in the rice *rab16B* promoter, and the binding of *OsBZ8* and *OsABI5* to ABRE was confirmed by a gel electrophoresis mobility shift and yeast one-hybrid assays, respectively. The presence of the cis-element DRE in the ABA-responsive gene *OsbZIP23* and of ABRE in the *OsNAC6* and *ZFP179* genes indicate a crosstalk between ABA-dependent and -independent signaling pathways [193]. In addition, the cis-element AH2 (CAAT (C/G) ATTG) interacts with the salt stress-related TF *Oshox22* to improve rice salt tolerance [194]. In addition to the various cis-acting elements identified above, similar-acting elements are found in the promoters of salt-responsive genes. For example, in rice, the promoter of the *C3HC4* ring finger gene contains cis-elements that are induced by salinity and other abiotic stresses [195]. At the same time, several cis-elements related to rice salt tolerance have been identified from different gene interaction networks [196]. All of these cis-elements may be involved in regulating salt-responsive gene expression.

In rice, several genes have been identified as having a role in sensing and transducing salt stress signals and an ability to enhance salt tolerance. Under high-salt conditions, overexpressing the protein kinase *SAPK4* can increase the seed germination rate, maintain intracellular ion balance, and improve photosynthesis and growth parameters by upregulating the H^+^-ATPase genes, *NHX1* and *OsCLC1* [96]. Overexpressing the calcium-dependent protein kinase *OsCPK12* resulted in enhanced salt tolerance due to reduced H_2_O_2_ accumulation in leaves and increased root biomass [197]. The MAPK gene *OsMAPK44* reduces plant damage induced by salt stress by maintaining a stable ion concentration in plant cells [97]. In addition, transgenic rice plants that were overexpressing another MAPK gene, *OsMAPK33,* displayed a reduced accumulation of harmful substances in cells and K^+^/Na^+^ ratio through the expression of ion transport-related genes or downstream salt-responsive positive regulators in the MAPK pathway [198]. *OsSRK1* encodes a typical S-type receptor kinase, and its overexpression induces the expression of *OsMyb4*, *OsDREB1A*, *ZOS3*, and *OsWRKY08* to improve rice salt tolerance [199]. Under salt stress, G protein, small G protein, and channel protein levels increase, triggering the expression of salt tolerance-related genes [200].

Transgenic plants overexpressing *OVP1* can increase the activities of tonoplast pyrophosphatase and ATPase, providing a proton driving force for the antiporter MHX and sequestering Na^+^ in the vacuole, reducing the damage of Na^+^ to the cytoplasm, and improving rice salt tolerance [201]. *SKC1* (*OsHKT8*), a member of the *OsHKT* family, is a Na^+^-specific transporter related to Na^+^ long-distance transport and regulates the above-ground Na^+^/K^+^ balance in rice under salt stress [202]. CDPKs directly bind to Ca^2+^, mediate calcium signal channels, and regulate ABA synthesis and response to salt stress [203]. Overexpressing *OsLOL5* increases rice salt tolerance and enhances the expressions of the oxidative stress-related genes *OsAPX2, OsCAT, OsCu/Zn-SOD*, and *OsRGRC2*, indicating that *OsLOL5* may enhance ROS scavenging to improve rice salt tolerance [98]. In rice, the zinc finger protein DST is downregulated under salt stress, resulting in the downregulated expression of ROS-scavenging genes such as catalase and peroxidase and leading to the accumulation of H_2_O_2_, stomatal closure, reduced water loss, and Na^+^ entrance, thus improving the plant’s salt stress tolerance [91].

Osmoprotectant accumulation is critical for alleviating the intracellular osmotic imbalance in plants under salt stress. In rice, the trehalose pathway genes, *OsTPP1* and *OsTPS1*, the proline biosynthesis pathway gene *OsP5CS*, and the glycine betaine biosynthesis gene *OsCMO* promote the overall growth rate of rice, attaining a high-salt tolerance [204,205]. Similarly, under salt stress, high light conditions and CO_2_ enrichment can enhance the expression of *OsBADH1* [99]. These two conditions can affect the photosynthetic efficiency of plants, and *OsBADH1* may protect the photosynthetic reaction from salt stress through various mechanisms. Therefore, the biosynthesis and expression of osmoprotectant-related genes are crucial for rice salt stress tolerance.

Homeostasis is essential in determining whether plants can overcome salt stress damage. For example, some protein transport receptors, such as ion pumps or ion channels, maintain the K^+^/Na^+^ ion balance in rice under salt stress. Overexpressing several ion balance-related genes, such as *OsKAT1*, *OsHAK5,* and *OsVP1*, can enhance salt tolerance, mainly by increasing the growth rate, photosynthesis, and root biomass. In addition, as a chloride channel-encoding gene, *OsCLC1* is upregulated under salt stress and maintains yeast growth under salt stress [100,101,102,206]. The decreased expression of *OsTPS1* results in Ca^2+^ deposition, limiting Na^+^ entry into the apoplast [207]. Therefore, three genes, *OsCNGC1*, *OsCAX,* and *OsTPS1*, may negatively regulate rice salt tolerance.

In addition, the genes associated with ROS are involved in regulating salt tolerance in rice. Transgenic rice overexpressing *OsECS*, *OsVTE1*, and *OsMSRA4.1* displayed high-salt tolerance [208,209]. Under salt stress, the exogenous expression of the rice dehydroascorbate reductase gene *DHAR* increases the germination and seedling growth rate of transgenic Arabidopsis by accumulating high levels of ascorbic acid [210]. In addition, overexpressing the cytoplasmic peroxidase genes, *OsAPXa* and *OsAPXb,* improves salt tolerance [211].

Under salt stress, the ectopic expression of the HSP genes *OsHsp17.0* and *OsHsp23.7* can improve cell membrane stability, germination ability, free proline content, and seedling survival rate while effectively preventing electrolyte leakage and reducing MDA content [212]. In addition, overexpressing the cyclophilin gene *OsCYP2* confers a high photochemical efficiency, reduced MDA levels, and increased antioxidant enzyme activity in rice under high-salt conditions, suggesting a key role for this gene in salt stress signaling pathways. The expression of *OsRab16A* under high-salinity conditions can lead to rapid rice growth, the accumulation of osmotic substances, enhanced antioxidant capacity, and the balance of trace elements [213]. In Arabidopsis, overexpressing *OsLEA3-2* increased the seed germination rate under salt stress, presumably acting as a molecular chaperone to maintain enzyme activity during dehydration and effectively preventing destructive protein aggregation [214]. In summary, HSPs and molecular chaperone genes can improve rice salt tolerance by maintaining the stability of protein properties and structures. Most importantly, the roles of sodium transporters in regulating ion homeostasis and the salt stress response require further attention.

### 2.5. Osmotic Stress

When plants are subjected to low temperatures, drought, and salinity, the concentration of the ion transport changes, leading to an increase in a cell’s permeability to water. This causes the accumulation of many small organic molecules, such as sugars, proline, and betaine, which have osmoprotective effects. Furthermore, there is an increase in the concentration of ABA, which triggers a series of physiological and biochemical reactions, leading to changes in the protein composition [215,216]. To maintain an osmotic pressure balance, plants regulate ion absorption and compartmentation, facilitate water intake, enhance antioxidant defense systems, and alter photosynthetic pathways while permeabilizing the biosynthesis of compatible solutes. Thus, some cellular responses arise from primary stress signals, while others arise primarily from secondary signals. Drought stress and salt stress have independent and some common signal transduction mechanisms. In addition, cold stress reduces the root’s water uptake in rice, but prolonged exposure to low root temperatures induces a gradual increase in the root’s osmotic hydraulic conductivity [217]. To avoid osmotic stress, plant cells reduce the intracellular water potential and maintain intracellular water through osmotic adjustment, thereby ensuring the normal physiological activities of plant cells. Substances involved in osmotic regulation are called osmotic regulators, mainly including inorganic ions, such as Na^+^, K ^+^, Cl^-^, and small organic molecules, such as polyols (mannose), nitrogen-containing compounds (proline and betaine), sugars, organic acids (malic acid), and their derivatives. Proline is the most water-soluble amino acid, and its functions include being an intracellular osmotic regulator, reducing agent, energy source, hydroxyl radical scavenger, intracellular enzyme protection agent, N-element storage substance, and regulator of redox potential. When rice seedlings are subjected to osmotic stress, the proline content in leaves and roots is significantly involved in cell osmotic regulation [218].

δ-Pyrroline-5-carboxylate synthase (P5CS) is the rate-limiting enzyme that catalyzes the first two steps in proline biosynthesis from glutamate, and is regulated by the transcriptional level of *P5CS* [219]. Overexpression of the gene *OsP5CS1* can significantly increase the proline content and improve the osmotic tolerance, thereby reducing the oxidative damage of rice under salt stress [103]. The wheat gene *Ta-UnP* regulates the expression of *P5CS,* and the overexpression of *Ta-UnP* in rice can highly induce the expression of *OsP5CS1*, resulting in increased proline content, and improving the osmotic and salt stress tolerance of rice [220]. The wheat gene *TaPUB15* encodes a U-box E3 ubiquitin ligase, which is highly expressed in seedling roots and induced by salt stress. The overexpression of wheat *TaPUB15* in rice can significantly increase the transcription level of *OsP5CS1*, and at the same time, the transgenic lines have more roots and an improved salt tolerance [221]. The Alfin-like (AL) family is a group of small plant-specific TFs involved in abiotic stresses in dicotyledons. In an early study, it was found that an *AL* gene in rice was associated with the grain yield under drought stress. It was also found that Hap1 of *OsAL7.1* and Hap7 of *OsAL11* were favorable haplotypes of seed weight and germination under osmotic stress. Additionally, during the germination stage, the *osal7.1* and *osal11* mutants have larger seeds and are more susceptible to mannitol and abscisic acid. In contrast, the overexpression of *OsAL7.1* and *OsAL11* reduced the stress tolerance at the adult stage [222]. Wang et al. [104] used the CRISPR/Cas9 system to obtain a mutant of the inorganic pyrophosphatase *OsPPa6* gene. In the mutant, inorganic phosphorus, ATP, chlorophyll, sucrose, starch, net photosynthetic rate, soluble sugar, and proline were significantly reduced, while the MDA, osmotic potential, and Na^+^/K^+^ ratio significantly increased, indicating that the *OsPPa6* gene is an important osmoregulatory factor in rice. Calcium-binding proteins are important in signal transduction for growth and stress response. In rice, OsCCD1, a novel small calcium-binding protein with a centrin-like domain, was characterized. OsCCD1 binds Ca^2+^, and its expression is induced by osmotic and salt stress and positively regulates tolerance to these stresses through involvement with genes such as *OsDREB2B*, *OsAPX1*, and *OsP5CS* (Figure 2B). It was observed that osmotic and salt stress dramatically increased the expression of *OsCCD1* via the calcium-mediated ABA signal [105].

Annexin is a multi-gene family of calcium-dependent phospholipid-binding proteins that are found in plants and other organisms. In plants, annexins play various roles such as membrane trafficking, abiotic stress responses, and signal transduction [223]. Interestingly, it was found that *OsANN10*, a putative annexin gene in rice, negatively regulated plant responses to osmotic stress. Knocking down *OsANN10* significantly decreased the content of H_2_O_2_ by increasing POD and CAT activities, suggesting a negative regulation of *OsANN10* in protecting the cell membrane against oxidative damage via scavenging ROS under osmotic stress [106]. Proteins that regulate cell wall polysaccharide synthesis play a crucial role in osmotic stress tolerance in plants. During osmotic stress, these proteins control the deposition and reorientation of cell wall components, which helps maintain cell integrity and prevent water loss. In a study, the role of the protein OsCSLD4, involved in regulating cell wall polysaccharide synthesis was evaluated in response to osmotic stress. The study shows that OsCSLD4 plays a positive role in osmotic stress tolerance in rice by regulating the ABA content. This study also conducted a transcriptomic analysis to investigate the genes involved in the response to osmotic stress and their regulation by OsCSLD4. The findings suggest that OsCSLD4 plays a crucial role in regulating the expression of genes involved in ABA biosynthesis and signaling pathways, which are essential for the response to osmotic stress [107].

The balance between hormonal signaling and osmotic stress tolerance in plants plays a crucial role in promoting plant growth and development and regulating stomatal closure and the accumulation of compatible solutes. It was found that the *OsGA2ox8* is induced by various abiotic stresses and phytohormones, and its overexpression enhances osmotic stress tolerance in rice by increasing osmotic regulators and antioxidants. It was observed that *OsGA20ox8* is preferentially expressed in shoots and roots under osmotic stress and is also involved in regulating genes that are associated with anthocyanin and flavonoid biosynthesis, as well as JA and ABA biosynthesis pathways [224]. Recently, *OsNF-YA3*, a rice TF, was found to regulate this GA and ABA balance by activating GA biosynthetic genes and enhancing GA content while repressing the ABA response to stabilize plant growth and osmotic stress tolerance. It was observed that *OsNF-YA3* negatively regulates plant osmotic stress tolerance by binding to the promoters of ABA catabolic genes (*OsABA8ox1* and *OsABA8ox3*) and reducing ABA levels. The activity of OsNF-YA3 is inhibited by the DELLA protein SLR1, and OSMOTIC STRESS/ABA-ACTIVATED PROTEIN KINASE 9 also interacts with *OsNF-YA3* and mediates its degradation [225]. The specific roles of type 2C protein phosphatases (PP2Cs) in rice abiotic stress tolerance are not well understood. In rice, 90 *PP2C* genes have been predicted, but very few members have been functionally characterized [226]. In a previous study, the function of *OsPP65* in osmotic and salt stress tolerance in rice was investigated. It was observed that *OsPP65* is expressed in rice seedlings and leaves and induced by multiple stresses. The knockout of *OsPP65* increased osmotic and salt stress tolerance in rice plants through independent regulation of JA and ABA signaling pathways. Metabolomics analysis indicated modulation of raffinose family oligosaccharide metabolism pathway in rice. *OsPP65* is a potential target for improving rice stress tolerance using gene editing [108].

In rice, the HSP90 family gene, *OsHSP50.2,* was found to be ubiquitously expressed, and its expression was induced by heat and osmotic stress treatments. The *OsHSP50.2* overexpression lines showed reduced water loss and enhanced or improved tolerance to osmotic and drought stresses. Overexpression lines exhibited significantly lower levels and less electrolyte leakage but higher SOD activity. It was also observed that the *OsHSP50.2*-overexpressing plants accumulated a significantly higher proline content and improved osmotic adjustment to drought stress damage [227]. In Arabidopsis, a hyperosmolality senser, *OSCA1* (*osmolality-sensing ion channel 1*), was identified. The homolog of *OSCA1* from rice (*OsOSCA1.2*), consisting of 11 transmembrane (TM) helices and a cytosolic soluble domain that has homology to RNA recognition proteins, also mediates hyperosmolality sensing, transport pathway gating, and balances the intracellular Ca^2+^ concentration. The TM domain is similar to the TMEM16 family and has a unique structure with extended helical arms. These arms may detect tension on the lipid bilayer caused by turgor pressure and open a transport pathway, thus enhancing osmotic stress tolerance [109]. Receptor-like cytoplasmic kinases (RLCKs) are mainly involved in growth regulation and pathogen responses in plants; however, their role in abiotic stress tolerance remains elusive. A study found that *OsRLCK241* is not only induced by salt and drought stresses but also by ROS detoxification by accumulating more compatible osmolytes and enhancing the activities of ROS scavengers to alleviate the osmotic stress evoked by drought and stress [228]. The rice plants have evolved various mechanisms to tolerate osmotic stress, including the regulation of gene expression, the synthesis of compatible solutes, and hormonal regulation. However, knowledge about the complex signaling pathways involved in osmotic stress responses in rice, as well as the interactions between different stress response pathways, is still very limited compared to other stresses.

### 2.6. Submergence Stress

Submergence or flooding can affect oxygen, CO_2_, light, and nutrient uptake, thus inhibiting photosynthesis, accelerating energy consumption, and causing plant growth atrophy or death [229,230]. To adapt to excess water conditions, rice has developed specialized anatomical and morphological traits, such as aerenchyma, radial oxygen loss barriers, adventitious roots, and the ability to form a leaf gas film. In addition, plants produce ventilated tissues and ethylene to facilitate gas exchanges and the programmed cell death of cortical and epidermal cells [231,232]. Moreover, the mechanical force generated by the growth of adventitious roots regulates epidermal cell death [233]. However, these strategies are insufficient for survival under continuous and complete submergence, resulting in stunted growth or complete death because of photosynthesis inhibition and the fast consumption of energy reserves. The submergence conditions rapidly trigger GA accumulation, resulting in rapid internode elongation [234]. When submerged, rice plants can form a protective wall around their roots called a radial oxygen loss (ROL) barrier from the root base to the root tip and outside the aerenchyma to prevent oxygen loss that has reached the roots [235]. Gene expression and mutant analyses suggested that suberin is the main component of the ROL barrier in rice roots, but the detailed mechanism of barrier formation remains to be clarified [236,237]. To overcome prolonged submergence, some Asian rice varieties have developed additional traits, such as aerobic germination, quiescence of leaf elongation in response to flash floods, and internode elongation under periodic flooding. Some rice cultivars tolerate submergence for ~15 days by restricting the consumption of carbohydrates, degradation of chlorophyll, and elongation growth [110,238].

Rice adaptation to submergence stress includes submergence escape and tolerance, which are controlled by *SNORKEL* (*SK*) and *SUBMERGENCE-1* (*Sub1*), respectively, belonging to the ETHYLENE RESPONSIVE FACTOR (ERF) class of TF genes [234,238]. The functional analysis of *ERF*-type TF genes revealed their involvement in controlling various physiological and morphological responses under submergence conditions.

When rice germinates under anaerobic conditions, radicle elongation is suppressed, while the coleoptile elongates and exposes its tip to the water surface to secure oxygen [239,240]. During anaerobic germination, the expression of the amylase alcohol dehydrogenase (ADH) genes is induced, leading to stored starch decomposition and ethanol fermentation to secure energy for elongation and promote anaerobic respiration. In rice, *OsTPP7* (*trehalose6-phosphate phosphatase 7*) is involved in anaerobic respiration, promoting energy supply to the coleoptile by increasing the metabolism of trehalose 6-phosphate (T6P) in the embryo [241].

Submergence-tolerant rice exhibits growth atrophy, which is beneficial to reduce energy consumption and ensure rice survival for several weeks under flooding stress and sufficient energy under continuous submergence. A major QTL (*Sub1*) on chromosome 9 of the submergence-tolerant *indica* cultivar ‘FR13A’ explains 70% of the phenotypic variation [242]. The *Sub1* region of submergence-tolerant varieties contains clusters of three similar genes, *SUB1A*, *SUB1B*, and *SUB1C*, encoding ERF-like TFs, among which *SUB1A* is the most studied. *SUB1B* and *SUB1C* are invariably present in the *Sub1* region of all the rice accessions analyzed. Comprehensive genetic studies revealed that *SUB1A* introgression with *SUB1B* and *SUB1C* confers a robust tolerance to submergence without affecting rice grain yield and quality [110,238,243,244]. In addition, *SUB1A* is present in a limited number of *indica* and *aus* cultivars, whereas *SUB1B* and *SUB1C* are found in all rice accessions [238]. *SUB1A* reduces energy consumption by negatively regulating the expression of genes that encode enzymes that degrade starch and sucrose and positively regulating ADH and pyruvate under submergence conditions [110]. In addition, *SUB1A* inhibits ethylene synthesis and the expression of cell wall loosening and expansion proteins under flooding stress, maintaining high chlorophyll a and b contents (Figure 2C). Through magnetic resonance, metabolites controlled by *SUB1A* under flooding stress were analyzed. As a result, *SUB1A* was involved in carbohydrate consumption, amino acid accumulation, and aerial part elongation [245]. In rice, two ERF multigenic *SUBMERGENCE-1* (*SUB1*) and *SNORKEL* (*SK*) loci govern the quiescence versus escape antithetical adaptive responses [110,234,238,246]. The detailed molecular and physiological studies of *SUB1A* and *SK* genes indicated that these *ERF VII* genes respond to submergence through the same hormonal pathways and regulate antithetical growth. *SUB1A* and *SKs* display higher transcriptional accumulation in response to ethylene under submerged conditions [110,234]. *SUB1A positively regulates adaptability to flash flooding, as plants are re-exposed to atmospheric oxygen and post-anoxic injury because of oxidative stress* [247]. In addition, *SUB1A* improves oxidative stress tolerance by regulating gene-encoding enzymes related to ROS detoxification [248]. Interestingly, *SUB1A* induces the transcription and protein accumulation of *SLR1* (*slender rice1*), a suppressor of GA signaling, causing elongation suppression in submerged conditions [246]. Rice plants that are sensitive to flooding lack the genomic region containing the *SUB1A* gene or have mutations that cause amino acid substitutions in the Sub1A protein [238]. The amino acid residue involved in this mutation is essential for the phosphorylation of *Sub1A* by MPK3, an MAP kinase [249]. Phosphorylated *Sub1A* induces the transcription of *MPK3*, forming a positive feedback loop, and the transcription of *ERF66* and *ERF67*. Those genes induce the transcription of genes that contribute to flood tolerance, such as *ADH* [250].

Two key genes, *SNORKEL1* and *SNORKEL2*, were cloned and studied. As a result, the study indicated that they respond to flooding stress by encoding response factors involved in ethylene signaling [251]. Under submergence, ethylene accumulates in rice and induces the expression of *SNORKEL1* and *SNORKEL2*, finally promoting internode elongation through GA. An attempt was made to identify the genetic factors for floating rice; they were identified on chromosomes 1, 3, and 12 [252,253,254,255]. A subsequent linkage analysis, using ‘Taichung No. 65’ (non-floating rice) and ‘C9285’ (floating rice from Bangladesh), revealed that the QTL located on chromosome 12 was the same ethylene responsive gene as *SUB1A*. *SNORKEL1* and *SNORKEL2* were identified in the genomic region deleted in non-floating rice. The OsEIL1 protein is stabilized in response to ethylene and regulates downstream gene expression [111]. In floating rice, the expression of *SNORKEL1*/*2* is induced by the OsEIL1 protein stabilized by ethylene accumulation under deep-water stimulation, thus promoting internode elongation through the expression of downstream genes. Interestingly, *SNORKEL1*/2 belongs to the same subgroup of the ERF family as *SUB1A* and *ERF66*/*67* [234,250]. Elucidating the detailed signaling mechanisms downstream of *SNORKEL1*/*2* may provide a molecular evolutionary answer to explain why similar TFs function differently in flood adaptation.

To search for new factors that are involved in floating rice, a GWAS was performed on 68 rice cultivars collected from various Asian countries with various floating rice qualities and different internodal lengths [256]. As a result, a region on chromosome 1 was strongly correlated with the internode length. A high-density linkage analysis was performed, and *GA20oxidase2* and *SD1* (*GA 20 oxidase 2*; *SEMIDWARF1*) involved in GA synthesis were identified in the region controlling internode length in floating rice. Comparing different high-floating rice cultivars revealed 17 polymorphic mutations in the promoter and intron regions commonly found in rice [257]. The *SD1* allele, *SD1DW,* containing these mutations significantly affected floating rice in the presence of *SNORKEL1*/*2*. *SD1DW* produces the active GA hormone, GA4, rather than GA1, which has a stronger effect on internode elongation. Later, it was found that *OsEIL1* directly binds to a specific region of the *SD1DW* promoter, revealing that SD1DW is under the control of *OsEIL1* and *SNORKEL1*/*2* [258].

Using GA-dependent internode elongation as an index, another gene, *ACCELERATOR OF INTERNODE ELONGATION 1* (*ACE1*), was identified as being induced by GA in a deep-water environment and ultimately inducing internode elongation. However, overexpressing *ACE1* alone does not result in internode elongation; initiating internode elongation in floating rice requires *ACE1* and independent GA signals. Another QTL, *DEC1,* encoding a C_2_H_2_ zinc finger-type TF and regulating GA-dependent internode elongation, was identified on chromosome 12. Overexpressing *DEC1* inhibited internode elongation, while *dec1* mutants displayed internode elongation. *DEC1* was expressed near the inter-meristem, where cell division was activated in the *dec1* mutant. In floating rice, GA administration and deep-water conditions decreased the expression of *DEC1*, suggesting that this decrease leads to the release of cell division inhibition during internode elongation. Five *ACE1-like* (*ACL*) sequences are found in rice, and overexpressing *ACL1* induces internode elongation when administered with GA. *ACE1* has lost its function in rice, while *ACL1* retained it; its expression is induced during reproductive growth [112].

## 3. Utilization of Integrative Omics Analyses to Identify Potential Candidate Genes Associated with Abiotic Stress

Recent advances in transcriptomic, proteomic, and metabolomic technologies now offer opportunities for candidate gene identification for subsequent utilization in plant breeding and potential improvements in global food security. The efficacy of breeding for increased environmental stress tolerance could be significantly improved through candidate gene identification and their utilization in marker-assisted selection. Several candidate genes underlying rice abiotic stress tolerance have been identified using functional genomics platforms. This section describes some examples from the literature about such candidate genes. Several stress responsive key genes, proteins, and metabolites have been uncovered through integrative omics studies and can be functionally characterized to utilize in breeding programs (Table 2).

### 3.1. Transcriptomic Data

The RNA-seq analyses upgraded our knowledge of gene expression in rice under different environmental conditions. Recently, RNA-seq has become more advanced, with increased detection sensitivity and sequencing depth. To date, transcriptome analysis has identified candidate genes linked to several abiotic stress responses in rice plants. A transcriptome study identified several DEGs for chilling stress, and by combining QTL fine-mapping and functional identification, the co-localized DEGs may provide insights for elucidating the mechanism of chilling tolerance and the basis for gene cloning. The study identified 154 highly expressed genes encoding oxidoreductases, thioredoxins, and glutathione S-transferases, among others. These genes help maintain cell redox homeostasis and promote chilling tolerance. Some stress-responsive genes encoding death-associated protein kinase 1, calcium homeostasis regulator CHoR1, four LRR-containing proteins, and four F-box domain-containing proteins were also part of this group. Another group of 27 genes, including 2 genes encoding short-chain dehydrogenase/reductase family proteins, 6 genes encoding TFs, and 3 stress-responsive genes encoding senescence-associated protein DIN1, polygalacturonase inhibitor 1, and hairpin-induced protein 1, were identified, indicating their positive role in chilling stress responsiveness. Several genes with a negative role in redox regulation, metabolism, stress response, and transport were identified, including genes encoding peroxidases, wall-associated kinase 3, oxidoreductases, and receptor-like kinase ARK1AS. Interestingly, two HSPs, an HSF, a photosystem II D2 protein, and a set of protein kinase domain-containing proteins were also discovered. Although HSPs and HSFs are known for their role in heat shock response in plants and other organisms, their function in chilling stress responsiveness in plants remains to be analyzed [259].

In an integrated transcriptome-based study of heat-resistant and -tolerant *indica* rice varieties, three candidate genes were reported as essential for high-temperature stress tolerance at the seedling stage. First, the study identified genes related to ROS (*OsCML4*), plant hormone signal transduction, metabolic pathways, cysteine and methionine metabolism pathways (*LOC_Os01g09450*, *LOC_Os03g59040*, and *LOC_Os12g42980*), stomatal conductance (*LOC_Os03g59040*), and survival rate (*LOC_Os02g12890*) under heat stress. Finally, *LOC_Os01g09450*, *LOC_Os03g59040,* and *LOC_Os12g42980* were potential candidate genes for high-temperature stress tolerance in seedlings [260]. Another study utilized GWAS and a transcriptome analysis and identified 11 genes that were associated with the rice heat stress response by examining the SNP variation and gene expression. LOC_Os03g16460 encodes a putative uncharacterized protein, and *LOC_Os05g07050* encodes a putative pre-mRNA processing splicing factor 8. By analyzing the 221 rice accessions, seven mutation sites classified into five categories and three mutation sites classified into two categories were found for *LOC_Os03g16460* and *LOC_Os05g07050, respectively*. Amino acid changes were observed in the encoded proteins due to these mutations. For *LOC_Os03g16460*, plants with mutations at all seven sites displayed the highest heat tolerance, whereas plants carrying a single mutation displayed a relatively low heat tolerance—a low survival rate. Similarly, for *LOC_Os03g16460*, plants with a single mutation were less tolerant to heat than plants with all three mutations. Interestingly, plants with more mutation sites displayed higher heat tolerance in *indica* rice accessions, whereas fewer mutation sites in *japonica* accessions tended to have lower heat tolerance [261].

Subsequently, several heat-responsive genes mainly involved in stress response, transport, transcriptional regulation, protein binding, and antioxidant activities were identified through transcriptomic analyses. The time-dependent expression pattern of genes under heat stress included several heat shock TFs (Hsf) and Hsp candidate genes. *Hsf (Os02g0527300, Os03g0133000,* and *Os06g0662200)*, *Hsp (Os03g0218500, Os03g0224700,* and Os04g0107900), glycolysis (*Os01g0160100*, *Os05g0187100*, and *Os05g0524400*), and ubiquitin–proteasome system (Os03g0116700, *Os05g0160200*, and *Os06g0173100*) gene expression was generally enhanced under heat stress, whereas genes related to specific secondary metabolisms (*Os06g0656500*, *Os08g0498400*, and *Os09g0492700*) and light reaction (*Os01g0720500*, *Os01g0938100*, and *Os07g0105600*) were repressed by heat stress [262]. It is worth mentioning that the whole transcriptome identified 37 novel heat-responsive genes, including many TFs. Remarkably, only 22 out of the 37 genes were heat-responsive and part of a tissue-independent general response; they were also identified in another RNA-seq study. In addition, 46 genes representing HSFs (*OsHSFA2f* and *HSFB2c*), molecular chaperones, and co-chaperone families were highly regulated during anthesis under heat stress. Interestingly, a small *HSP* (*LOC_Os02g52150*) was also highly upregulated in the transcriptomic data [263]. Another transcriptomic investigation suggested 589 candidate genes related to heat stress responses in rice seedlings. Six heat stress-responsive genes were identified that are associated with a broad range of abiotic stress responses, such as drought, salt, cold, and submergence. These genes are potential candidates for fighting combined stresses. In addition, *OsNAC6*/*SNAC2* (*Os03g60080*) was upregulated under heat and drought stress, indicating a strong correlation between the two. Furthermore, five *HSP10*/*CPN10* and thirteen *HSP20* genes were associated with prolonged heat stress responses, but none of the *HSP10*/*CPN10* or *HSP20* genes were associated with an early heat response. Among them, eight *HSP20* genes were significantly upregulated in response to multiple abiotic stresses, especially heat and drought. On the other hand, *HSP10*/*CPN10* were not stimulated by other abiotic stresses. These findings suggest that the HSP20 family proteins have evolved to cope with multiple abiotic stress challenges and prolonged heat stress, whereas the *HSP10*/*CPN10* family is specialized to overcome prolonged heat stress [264]. In addition to heat stress, several transcriptome studies have been utilized and successfully identified candidate genes involved in other stresses.

Another transcriptome analysis was performed on 4-week-old rice seedling roots subjected to drought stress for two and three days. As a result, 1098 genes were upregulated in response to 3-day drought stress, which caused severe damage to root development after recovery, unlike the 2-day stress. Interestingly, 68% of candidate genes were not identified in previous transcriptomic studies. Further bioinformatics analysis revealed that RING-box E3 ligases in the ubiquitin–proteasome pathways are significantly stimulated by drought. Finally, the study analyzed the functions of 66 candidate genes and found 29 genes directly involved in drought tolerance. The identified genes were involved in metabolic pathways, protein modification, and protein degradation [265]. In the transcriptomic study, several *NAC* (CUC, NAM, and ATAF) family genes were upregulated under salt stress, and among them, 14 were reported in previous studies to be associated with abiotic stress responses. Additionally, two *HD-ZIP* and two *MYB* genes were highly upregulated, making them candidates for salt stress tolerance in rice [266]. Under salt stress, a transcriptome study of rice roots found that several candidate genes related to multiple stresses. A further analysis identified 13 candidate genes that were already functionally characterized and involved in phytohormone signaling and salt stress tolerance [267].

Another study identified non-coding RNA—miRNAs and lncRNAs—and several candidate genes involved in salinity and submergence stress responses through comparative transcriptome analyses of wild rice under different stress conditions [268]. RNA sequencing of two contrasting recombinant inbred lines (RILs) retrieved five candidate genes, including *LOC_Os01g04430*, *LOC_Os01g04530*, and *LOC_Os03g22720* for *qCL*-*1.1* and *qCL*-*3.1*. These genes are responsible for different metabolic processes, and the RILs expressing these genes were submergence-tolerant [269]. After utilizing the deep transcriptomes of rice leaves and roots under drought, cold, and salt stresses, many upregulated and downregulated genes specifically and commonly expressed under those stress conditions were identified. Data mining identified the expression of key functional and regulatory genes and conserved cis-regulatory elements in the promoter of highly presented genes involved in different abiotic stress signaling pathways [270].

Although several transcriptome studies have reported candidate genes and TFs involved in the response to abiotic stress in rice, the expression landscape of these genes at a single-cell resolution remains poorly understood. From the perspective of transcriptomic work, the importance of individual genes and gene sequences can be further confirmed in genome editing experiments.

### 3.2. Proteomics Data

A proteomics analysis provides a broad picture of plant stress responses at the protein level. Because of advancements in two-dimensional polyacrylamide gel electrophoresis (2DE), protein detection and quantification, fingerprinting, and partial sequencing by mass spectrometry, the sensitivity, and power of proteomic approaches have increased in recent years.

Some candidate components, including a protein similar to WAK1, a putative armadillo repeat-containing protein, and a protein phosphatase 2C-like protein involved in signal transduction under cold stress—RAB2A—were identified [271]. In a proteomics analysis of cold-treated anthers, 441 DEPs were identified, and following a bioinformatics analysis, 30 upregulated and 4 downregulated were considered candidate genes. Most were related to glycine-rich proteins and C2 domain proteins [272]. An iTRAQ-based proteomics approach identified 85 proteins related to low temperature stress, enriched mainly in transport, photosynthesis, precursor metabolism, and energy production, as well as histones and vitamin B [273].

The proteomic analysis of rice under drought stress alone and In combination with heat stress identified proteins favoring pollen germination. A comparative iTRAQ-based proteomics study compared the proteomes of heat-tolerant and -sensitive rice lines at an early milky stage. A total of 38 DEPs were identified, among which 32 were functionally annotated in NCBI and UniProt databases. These proteins were related to the defense response, transport, energy metabolism, signal transduction, transcript regulation, oxidation, and biosynthesis. A further analysis identified downstream genes/proteins involved in the high light temperature response [274]. Another proteomics study identified 54 candidate proteins involved in high-temperature stress during the first half of the ripening period of heat-sensitive rice lines [275].

Furthermore, two small HSPs (16.9 and 17.4 kDa) were strongly expressed under combined stress at the protein and transcript levels. Comparing the protein and transcript expression patterns for these two genes suggested differential gene regulation at the transcriptional and translational phase in response to heat and drought stress [276]. Proteomics analysis was performed to elucidate the molecular responses of Thai jasmine rice under drought, and 623 DEPs were identified. After a comprehensive analysis, a candidate protein, DEAD/DEAH-box RNA helicase, was identified. It is involved in DNA repair and was reported as a novel protein for promoting plant stress tolerance [277,278]. In another work, iTRAQ-based proteomics identified 1221 DEPs in rice grown under control and light conditions, and a candidate gene-encoding rice β subunit of glyceraldehyde-3-phosphate dehydrogenase (*OsGAPB*) was identified and functionally characterized [279].

In a proteomic study, the response of rice suspension cells to salt stress using the iTRAQ technique was studied, and significant changes in the carbohydrate and energy metabolism pathways, redox signaling pathways, auxin pathways, and biosynthetic pathways of osmolytes were observed [280]. Another study used iTRAQ techniques to study the proteome changes in rice under salt stress and found 56 proteins showing significantly altered expression, 16 of which were mainly enriched in the processes of photosynthesis, anti-oxidation, and oxidative phosphorylation, which play an important role in maintaining energy balance and generating active oxygen under stress conditions [281]. In another proteomic study, the rice seedlings of spotted leaf 1 (spl1) were completely submerged for five days to initiate cell death and then exposed to air, and the samples were collected at different time points. Finally, PBZ1, as a putative cell death marker, was identified, which was highly inducible in spl1 [282]. In short, using proteomics to identify candidate proteins—and genes—will help improve rice abiotic stress tolerance, which is reflected in rice yield and quality.

### 3.3. Metabolomics Data

Metabolic compositional variations reflect the changes in gene expression occurring in stress responses. Studying metabolic pathways and their interconnections in the context of systems biology is progressively becoming common for identifying candidate genes. To reveal the candidate genes, integrating transcriptomics, metabolomics, and proteomics, the data provide a comprehensive understanding of the molecular mechanisms underlying stress tolerance in rice. Stress tolerance in rice requires a fully functional metabolism throughout the stress, but the regulation of and the link between associated metabolites and transcripts remain largely unknown.

The comparative metabolomics analysis was performed for two subspecies of rice with significant divergence in chilling tolerance. Finally, 9 metabolites (glutathione, putrescine, asparagine, β-Alanine, γ-Glutamylleucine, oxalate, mannose-6-phosphate, isocitrate, and nicotinamide mononucleotide) with obvious changes in relative abundance under chilling treatment [283]. The metabolomics analysis analyzed heat-treated single mature pollen grains in two contrasting rice spikelet fertility studies under heat stress. Finally, contrasting varietal differences in phosphatidylinositol (PI) (34:3) were detected in mature pollen, together with other 106 metabolites. The major categories were linked to lipid-related metabolites, organic acids, amino acids, and related metabolites, carbohydrates, cell wall-related metabolites, plant hormones, and cluster ions [284].

In rice, integrative transcriptomics and metabolomics analyses of drought-tolerant and susceptible cultivars explored transcript and metabolic responses under drought stress. A study found 4059 and 2677 DEGs and 69 and 47 differential metabolites under drought and normal conditions, respectively. A correlation analysis suggested several candidate genes based on DEGs involved in the associated metabolites’ pathways. The upregulation of 4-hydroxycinnamic acid and ferulic acid correlated with the performance of photosynthesis-related DEGs during the early stages of drought. The identified metabolites, 4-hydroxycinnamic acid and ferulic acid, were reported to be critical for rice drought tolerance, and the DEGs involved in the pathways of these metabolites were suggested as promising candidate genes for improving drought tolerance [285]. The potential 46 candidate genes identified from a transcriptomics study were confirmed through metabolite profiling in 21 cultivars under drought conditions. The study revealed that, during drought stress, the levels of most metabolites had a negative correlation with the performance parameters. However, eight metabolites showed a positive correlation, including allantoin, galactaric acid, gluconic acid, glucose, salicylic acid glucopyranoside, and three unidentified metabolites. Furthermore, 28 genes were found to have a significant correlation between expression level and performance under drought, with predominantly negative associations. Among those showing a significant positive correlation was a gene responsible for encoding a cytosolic fructose-1,6-bisphosphatase enzyme, which plays a key role in regulating C-metabolism [286]. Another transcriptome and metabolome analysis of two contrasting genotypes under salt stress revealed the repression of genes involved in metabolite degradation and the upregulation of genes involved in several pathways underlying salt tolerance. Combined analyses identified 43 candidate genes related to salt stress. With more precise analysis, the metabolic pathways leading to the primary metabolites examined were investigated, revealing two significant findings. Firstly, the accumulation of metabolites induced by salt stress in the shots of both genotypes was primarily limited to a few sugars, including glucose, fructose, and fructofuranans, as well as proline, all of which were ABA-mediated. On the other hand, salt-induced metabolic accumulation in the roots of both genotypes involved products of N metabolism, such as allantoin, urea, and glutamine, and trehalose was the only sugar with a relatively high accumulation in the roots of both genotypes. Finally, changes in gene expression led to the coordinated accumulation of key primary metabolites in the shoots or roots of the salt-sensitive rice line and salt-tolerant progeny, with 12 key genes identified that are related to the identified primary metabolites [287].

Metabolic profiling demonstrated the presence of *SUB1A* in the crossbred line M202 (*Sub1*), which has a higher tolerance than wild-type M202 under deep flood conditions. The presence of *SUB1A* was found to be associated with a reduction in carbohydrate metabolism in shoot tissue, which is consistent with its role in limiting starch catalysts for elongation growth. On the other hand, the absence of *SUB1A* resulted in an increase in sucrose consumption and the accumulation of amino acids synthesized from glycolysis intermediates and pyruvate. Alanine, produced through pyruvate metabolism, showed the largest difference between the two contrasting varieties under submergence conditions. However, elevated levels of glutamine, glutamate, leucine, isoleucine, threonine, and valine were also higher in the absence of *SUB1A*. Additionally, the compound alanylglycine (AlaGly) was identified and characterized, and its levels under submergence conditions were assessed [288]. In most cases, metabolomics has been utilized to study the expression of metabolites in response to different stresses, but very few studies have focused on identifying candidate genes. However, it is clear from the abovementioned studies that combining metabolomics with other omics approaches may enable the identification of the intended candidate genes.

**Table 2 plants-12-02019-t002:** List of some abiotic stress related key genes, proteins, and metabolites identified through omics studies.

Stress Type	No. of Key Factors	Functional Categories or Names of Factors	References
**Transcriptomics**			
Cold	154	Oxidoreductases, thioredoxins, glutathione S-transferases, and cell redox homeostasis	[259]
Heat	3	Plant hormone signal transduction, metabolic pathways, cysteine, and methionine metabolism pathways	[260]
	11	Pre-mRNA processing splicing	[261]
	18	Transcriptional regulation, transport, protein binding, antioxidant, and stress response	[262]
	22	Heat response, molecular chaperone, and co-chaperone	[263]
Drought	29	Metabolic pathways, protein modification, and protein degradation	[265]
salt	13	Phytohormone signaling and salt stress	[267]
Submergence	5	Metabolic processes	[269]
**Proteomics**			
Cold stress	34	Glycine-rich proteins and C2 domain proteins	[272]
	85	Transport, photosynthesis, precursor metabolism, and energy production, as well as histones and vitamin	[273]
Heat	32	Defense response, transport, energy metabolism, signal transduction, transcript regulation, oxidation, etc.	[274]
Drought	2	Heat shock proteins	[276]
	1	DNA repair	[277]
Salt	58	Carbohydrate and energy metabolism pathways, redox signaling pathways, auxin pathways, etc.	[280]
	16	Photosynthesis, anti-oxidation, and oxidative phosphorylation	[281]
Submergence	1	Programmed cell death	[282]
**Metabolomics**			
Cold	9	Glutathione, putrescine, asparagine, β-Alanine, γ-Glutamylleucine, oxalate, mannose-6-phosphate, etc.	[283]
Heat	109	Lipid-related metabolites, organic acids, amino acids and related metabolites, carbohydrates, etc.	[284]
Drought	2	Metabolites, 4-hydroxycinnamic acid, and ferulic acid	[285]
	8	Allantoin, galactaric and gluconic acid, glucose and salicylic acid glucopyranoside, etc.	[286]
Salt	16	Fructofuranose, fructose, glucose, proline, urea, and allantoin	[287]
Submergence	1	Alanylglycine	[288]

## 4. Conclusions and Perspectives

The molecular mechanism of response to abiotic stress has gradually become more precise in rice. In recent years, researchers have conducted many studies on the signal transduction and gene expression regulatory pathways during the response to major abiotic stresses, and many hormones’ and TFs’ roles in plant stress tolerance have also been revealed. However, most of the research progress at this stage focuses on discovering unknown TFs, editing promoter binding sites, and working on the biochemical level. New research directions, tools, and practices must be developed. The current research has identified many TFs that bind to cis-acting elements of target gene promoters to activate the expression of specific downstream genes. In rice, the binding sites of these TFs have been analyzed, and their binding mechanism has been studied in detail. However, further research on multiple TFs synergistically acting as negative feedback regulators during stress signaling is necessary. In addition, the research conducted on the exploration of alternative approaches such as epigenetic modifications or small RNA regulations is still very limited. So far, many studies have shown that epigenetic regulation is involved in the response process of plant abiotic stress, and this new understanding has also introduced new research content and directions for epigenetic research. Epigenetic regulation under abiotic stress is a dynamic process, so it is very difficult to study the epigenetic mechanisms of plants under stress. There are still research gaps to reveal the outcomes of epimutations of cloned genes and their regulatory networks related to abiotic stress responses and epigenetic information. The epigenetic modifiers and specific epigenetic modification sites of stress responses and the corresponding stress-response target genes have not been fully understood. Under abiotic stress, the interaction between epigenetic regulation and plant endogenous hormones at different levels needs to be further studied. A series of epigenetic regulatory factors have been identified to participate in plant abiotic stress responses; how to apply them to crop breeding and improve agricultural production needs to be further explored.

Big data analysis and a variety of new methods are helpful to quickly find key players for broad-spectrum stress tolerance and can be further cloned to study their regulatory mechanisms. Analysis of the structure of key proteins, especially receptor proteins, and identification of important protein domains and key amino acid sites facilitate target designs for gene editing and other technical means. In addition, revealing the mechanism of plant abiotic stress responses requires comprehensively utilizing genomics, transcriptomics, and proteomics methods at the molecular level. Identifying a novel gene network that can sense stresses, transduce signals, and activate TFs and downstream genes directly related to stress tolerance will effectively support conventional and molecular breeding programs and genetic engineering strategies. Co-expression networks are an attractive framework for gene interaction analysis and offer various applications, from functional gene annotation to the comparison of co-expression networks across species. This approach will further contribute to the elucidation of important biological processes and provide a valuable predictive tool for modern molecular breeding and crop engineering strategies. At the cytological level, with the rapid development of microscopy techniques, cryo-transmission electron microscopy or high-resolution fluorescence microscopy to observe changes in plant cell membranes, organelles, and cytoskeletons will help better understand and analyze the cellular responses to different abiotic stresses. In short, cutting-edge technologies, such as whole genome sequencing platforms, high-throughput integrative omics techniques and resources, targeted genome editing technologies, and synthetic techniques, are growing and require utilizing basic research in advanced breeding methodologies. These latest improvements in precision breeding and data analytics, combined with stress-tolerant plants, can produce rice for the global market and achieve global food targets.

## Figures and Tables

**Figure 1 plants-12-02019-f001:**
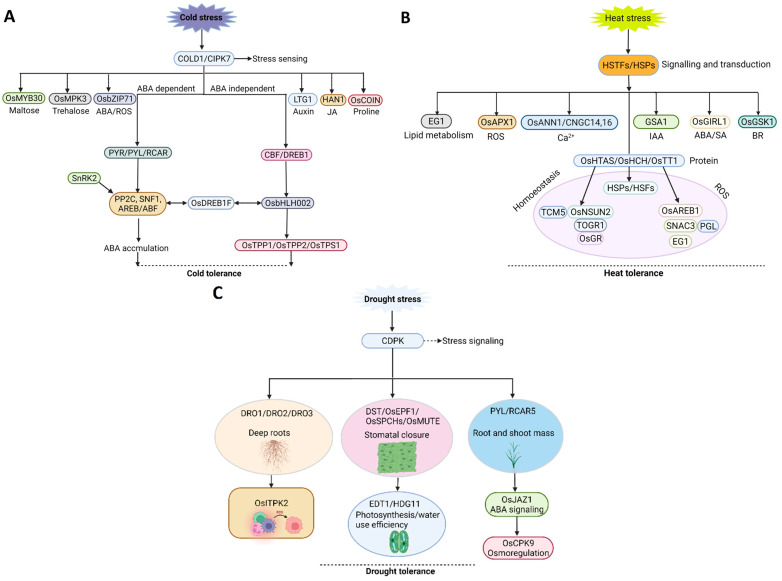
Mechanism of cold, heat, and drought stress sensing and tolerance in rice. (**A**) Cold stress mechanism and response. Cold stress signals are perceived by *COLD1* and *CIPK,* and different genes related to phytohormones and osmoprotectants are regulated. The upregulation of ABA-responsive genes leads to ABA accumulation and cold tolerance. (**B**) Heat stress sensing and response. Heat stress signals are perceived by different heat shock transcription factors and proteins. Different genes associated with ROS, lipid metabolism, Ca^2+^ homeostasis, and phytohormones are regulated. Several ROS and cell homeostasis genes are activated to trigger the heat stress response. (**C**) Pathway of drought sensing and tolerance. The root system is crucial for drought tolerance, and *DRO1* is upregulated under drought stress, leading to deeper roots and improved drought tolerance. Other genes associated with phytohormones, stomatal balance, water-use efficiency, osmotic adjustment, and root and shoot biomass are crucial for drought tolerance.

**Figure 2 plants-12-02019-f002:**
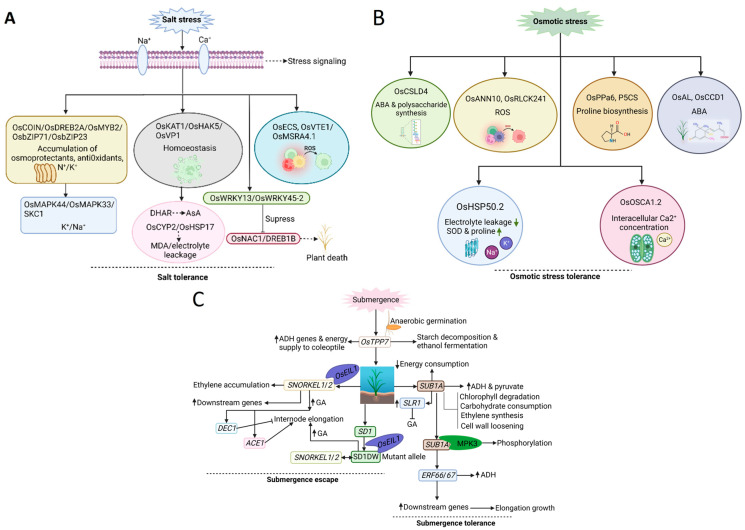
Salt stress, osmotic stress, and submergence stress tolerance mechanisms in rice. (**A**) Salt stress sensing and response. Different proteins play a role in antioxidant and osmoprotectant accumulation, ROS and Na^+^ homeostasis, MDA accumulation, and electrolyte leakage and are required to trigger the tolerance mechanism. Some WRKY TFs suppress the expression of *OsNAC1* and *DREB1B*, resulting in salt susceptibility. (**B**) Osmotic stress tolerance mechanism overview in rice. The genes involving ABA, proline, and polysaccharides biosynthesis are mentioned. Further, genes involved in ROS scavenging, preventing electrolyte leakage, and balancing the intracellular Ca^2+^ concentration are also highlighted. (**C**) Submergence tolerance response and regulatory mechanism. Rice follows the quiescent strategy to adapt and escape periodic flash flooding. *SUB1A* is the key regulator for submergence tolerance. It triggers the transcriptional regulation of *SLR1* and other ERF response factors. In floating rice, ethylene accumulation under deep-water conditions stabilizes the ethylene signaling factor *OsEIL1*. *OsEIL1* increases gene expression by binding to the promoter of *SD1.* After that, the accumulated gibberellin increases the expression of *ACE1;* meanwhile, the expression of *DEC1*, a factor inhibiting internode elongation, is reduced. On the other hand, OsEIL1 also binds to the promoter regions of *SNORKEL1* and *SNORKEL2*, triggering the expression of other downstream genes.

## Data Availability

Not applicable.

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
