# Peer review of "Recent Molecular Aspects and Integrated Omics Strategies for Understanding the Abiotic Stress Tolerance of Rice"

_plants, 2023, doi:10.3390/plants12102019_

Round 1
Reviewer 1 Report
The manuscript provides important information on abiotic stress factors limiting rice cultivation and molecular and omics approaches used for resilience of plants. The manuscript covers different aspects on abiotic stress as a general factor, and how there is a recognition and signaling of abiotic stress. The parts dealing with the transcriptome and proteomics studies are well addressed but other parts need revision and also more coverage on abiotic stresses that are not included. Authors should revise the manuscript to include major work done on rice in the addressed area in the last decade. Most studies referred are for general account of abiotic stress or other general information. Authors should try to collect information on signaling, transcriptomics, proteomics and metabolomics that confers tolerance in rice plants. Moreover, nutrient stress, the limitation and excess, should also be covered in the review. Similarly, work on osmotic stress should be addressed. In the beginning of the review more information on data on rice cultivation under different abiotic stress conditions, such area of land and yield and the expected decrease should be provided. It is better to exclude specific studies while describing general aspects of stress recognition as it is not possible to include all stresses and the examples here. Second para of heading 2.0 is well written. Under heat stress, soluble and non-soluble sugars with their relation to photosynthesis is important. Authors should include studies related to this aspect, and how high accumulation of sugars inhibits photosynthesis and the means to overcome should be discussed. In metabolomics, name the metabolomes covered in the studies with their significance. Conclusion: Please shorten the discussion here and include only major outcome of the review that may be taken as message, and the future prospects should cover the gap where more elaborate work has to be done.
The manuscript is written in good scientific language, only minor spelling errors are to addressed.
Author Response
The manuscript provides important information on abiotic stress factors limiting rice cultivation and molecular and omics approaches used for resilience of plants. The manuscript covers different aspects on abiotic stress as a general factor, and how there is a recognition and signaling of abiotic stress. The parts dealing with the transcriptome and proteomics studies are well addressed but other parts need revision and also more coverage on abiotic stresses that are not included.
We would like to thank the reviewer for the comments and nice suggestions to improve the manuscript. We believe that the comments have identified important areas that required improvement. After completion of the suggested edits, the revised manuscript has benefitted from an improvement in the overall presentation and clarity. We have tried to revise the whole manuscript and removed mistakes that we came across. Below, you will find a point-by-point description of how each comment was addressed in the manuscript. The response to comments is highlighted in red.
Comment 1: Authors should revise the manuscript to include major work done on rice in the addressed area in the last decade. Most studies referred are for general account of abiotic stress or other general information. Authors should try to collect information on signaling, transcriptomics, proteomics and metabolomics that confers tolerance in rice plants. Moreover, nutrient stress, the limitation and excess, should also be covered in the review. Similarly, work on osmotic stress should be addressed.
Response 1: Thank you for your valuable comments. We appreciate your suggestion and have revised the manuscript accordingly. In addition, we also aimed to include some general aspects combined with relevant studies that provide specific insights into abiotic stress in rice.
In addition, we have added a few more information about abiotic stress signaling identified through transcriptomics, proteomics and metabolomics studies (#section 3.1, 3.2. 3.3). We have carefully considered your suggestion to include information on nutrient stress and osmotic stress in the review. While we agree that these are important topics, our focus in this review is primarily on the major biotic stresses affecting rice growth and productivity. Therefore, we have chosen to prioritize the discussion of these specific stressors and their impact on rice plants. However, we have taken your feedback into account and expanded our coverage on osmotic stress (#section 2.5.) by including additional studies that provide insights into some key regulators. We hope that this will help to address your concerns while still maintaining our primary focus on the major abiotic stresses affecting rice production. Regarding nutrient stress, we have decided not to include it in this particular review as we believe it deserves a dedicated review to provide a comprehensive overview of its impact on rice growth and productivity. We look forward to exploring this topic in future research. Once again, thank you for your valuable feedback, and we hope that these revisions have improved the overall quality and relevance of our review.
Comment 2: In the beginning of the review more information on data on rice cultivation under different abiotic stress conditions, such area of land and yield and the expected decrease should be provided. It is better to exclude specific studies while describing general aspects of stress recognition as it is not possible to include all stresses and the examples here.
Response 2: Thank you for your constructive feedback. We appreciate your suggestion to include more information on data related to rice cultivation under different abiotic stress conditions such as the area of land, yields, and expected decrease We have revised the introduction section of our review to include more details on these aspects and provide a clear context for the discussion of biotic stresses affecting rice production. L23-62.
Comment 3: Second para of heading 2.0 is well written. Under heat stress, soluble and non-soluble sugars with their relation to photosynthesis is important. Authors should include studies related to this aspect, and how high accumulation of sugars inhibits photosynthesis and the means to overcome should be discussed.
Response 3: Thank you for your valuable feedback. We agree that the relationship between sugars and photosynthesis under heat stress is an important aspect of rice plant growth and grain development that merits discussion in our review. We have included a few additional studies that explore the impact of high sugar accumulation on photosynthesis under heat stress conditions (#Section 2.2, L328-354). We thank you again for your insightful feedback and for helping us improve the quality and relevance of our review.
Comment 4: In metabolomics, name the metabolomes covered in the studies with their significance.
Response 4: We have revised the manuscript to include more names of metabolites covered in this review.
Comment 5: Conclusion: Please shorten the discussion here and include only major outcome of the review that may be taken as message, and the future prospects should cover the gap where more elaborate work has to be done.
Response 5: We have revised the conclusion section and removed the redundant sentences. Moreover, we have a few sentences addressing the future prospects and research gaps. Thanks for the suggestion.
We have checked the whole manuscript again and made all possible corrections, including grammatical, sentence, words, and typo mistakes. We hope that this version of the manuscript is reader friendly. Thanks for the suggestions and for reading our manuscript carefully.
Reviewer 2 Report
The review of Usman et al. is a well-written manuscript. In recent years, the molecular aspects have been studied by numerous scientists majoring in different models and crop plants and resulting in an exponentially increasing of research articles and reviews published, while summarizing the integrated multi-omics strategies for abiotic stress responses of rice is generally interesting due to urgent need for sustainable production. I list some minor issues that can be considered by the authors.
- Line 32-33: more information on "various aspects" and "promotes social stability" can be given here for readers to get a big picture of how crucial it is to secure global demand.
- Fig. 1. Please consider specifying the differences between solid and dash lines and the whether there is any meaning about color codes between color blocks. For Fig. 1B, what is the meaning of "protein" besides OsHTAS/OsHCH/OsTT1? In Fig. 1C, What is the icon inside the block of OsITPK2?
- Fig. 2. The meaning of icons in Fig. 2A is not clear to me, the same comments as for Fig. 1, please consider explaining the meaning of different lines and colors of blocks.
- For Section 3, the multiple levels of omics data is one of the strengths of this manuscript but descriptive results for many publications can be difficult for readers to follow. Please consider summing up transcriptomics, proteomics, and metabolomics with table(s).
Author Response
RESPONSE TO REVIEWER #2
The review of Usman et al. is a well-written manuscript. In recent years, the molecular aspects have been studied by numerous scientists majoring in different models and crop plants and resulting in an exponentially increasing of research articles and reviews published, while summarizing the integrated multi-omics strategies for abiotic stress responses of rice is generally interesting due to urgent need for sustainable production. I list some minor issues that can be considered by the authors.
Thank you for your valuable feedback and time in reviewing our manuscript. Your input has been greatly appreciated and will undoubtedly enhance the quality of the review paper. We have carefully considered all your feedback and made the necessary modifications to the manuscript based on your recommendations.
Comment 1: Line 32-33: more information on "various aspects" and "promotes social stability" can be given here for readers to get a big picture of how crucial it is to secure global demand.
Response 1: We have added a few lines as per suggestions.
Comment 2: Fig. 1. Please consider specifying the differences between solid and dash lines and whether there is any meaning about color codes between color blocks. For Fig. 1B, what is the meaning of "protein" besides OsHTAS/OsHCH/OsTT1? In Fig. 1C, What is the icon inside the block of OsITPK2?
Response 2: We have modified the figure and removed the dash lines. Moreover, there is no meaning of colors in the figures. We have removed the word “protein” from Fig 1B. The icon inside the block is just to represent the ROS.
Comment 3: Fig. 2. The meaning of icons in Fig. 2A is not clear to me, the same comments as for Fig. 1, please consider explaining the meaning of different lines and colors of blocks.
Response 3: The icons are just inserted for the corresponding words. e.g., ROS, ABA and proline chemical formulas etc. These are just to enhance the beauty of figures and make them little reader friendly. Colors have no specific meaning here. The different colors are just to differentiate different genes. It will attract and help readers to understand the figures in an easy way.
Comment 4: For Section 3, the multiple levels of omics data is one of the strengths of this manuscript but descriptive results for many publications can be difficult for readers to follow. Please consider summing up transcriptomics, proteomics, and metabolomics with table(s).
Response 4: We have inserted a table. Thanks for the suggestions.
Reviewer 3 Report
I have been pleasantly surprised by this manuscript. It is very well written and covers the topic, at least in my opinion, very thoroughly.
A few minor items:
Line 61, should probably be "to" before "developing".
Line 81, "outer cell membrane" should probably be "plasma membrane".
In general, I would not abbreviate "ethylene" as ETH. A simple "E" would be okay. ETH is the official abbreviation of the Swiss University in Zurich and stands for "Eidgenössische Technische Hochschule Zürich".
There is also some redundancy in the text. For example, in the section on salt stress the interactions between Na and K should be explained once so it doesn't have to be repeated in several sections. I would recommend to check for this kind of repetition throughout the text and make the manuscript more compact (and may a little bit shorter).
Line 65,66; I would put "molecular" in front of "morphological". Every morphological change starts on the molecular level.
In the low temperature chapter, it should also be noted that ROS are important signaling compounds in the process and are not just damaging compounds.
As for the Heat Stress chapter, I would recommend toning down the effects of global warming a little bit. What we are experiencing are changes in weather in certain areas, but global warming per se does not cause extreme heat in general. Plants can easily tolerate a few degrees warmer or colder, but it's the changes in weather patterns. And while these can be caused by global warming, the overall changes in average temperatures are not really the cause of stress for plants.
Author Response
I have been pleasantly surprised by this manuscript. It is very well written and covers the topic, at least in my opinion, very thoroughly.
Thank you for reviewing our manuscript and providing insightful feedback and positive feedback. Your comments were beneficial in helping us to refine our manuscript.
A few minor items:
Comment1: Line 61, should probably be "to" before "developing".
Response 1: Correction has been made.
Comment 2: Line 81, "outer cell membrane" should probably be "plasma membrane".
Response 2: Correction has been made.
Comment 3: In general, I would not abbreviate "ethylene" as ETH. A simple "E" would be okay. ETH is the official abbreviation of the Swiss University in Zurich and stands for "Eidgenössische Technische Hochschule Zürich".
Response 3: Correction has been made. We used full form “ethylene” throughout the manuscript.
Comment 4: There is also some redundancy in the text. For example, in the section on salt stress the interactions between Na and K should be explained once so it doesn't have to be repeated in several sections. I would recommend to check for this kind of repetition throughout the text and make the manuscript more compact (and may a little bit shorter).
Response 4: We have removed redundant sentences and made possible changes to the whole manuscript. Thanks for the suggestions.
Comment 5: Line 65,66; I would put "molecular" in front of "morphological". Every morphological change starts on the molecular level.
Response 5: We moved the “molecular” before “morphological”. Thanks for the suggestion.
Comment 6: In the low temperature chapter, it should also be noted that ROS are important signaling compounds in the process and are not just damaging compounds.
Response 6: Thanks for point it. We have added necessary information.
Comment 7: As for the Heat Stress chapter, I would recommend toning down the effects of global warming a little bit. What we are experiencing are changes in weather in certain areas, but global warming per se does not cause extreme heat in general. Plants can easily tolerate a few degrees warmer or colder, but it's the changes in weather patterns. And while these can be caused by global warming, the overall changes in average temperatures are not really the cause of stress for plants.
Response 7: We have revised the first paragraph of section 2.2. We agree with reviewers that global warming is not affecting the gene expression differences in a specific manner. So, we have added a few sentences to make it clearer. Thanks for the suggestion.
Reviewer 4 Report
The authors' comprehensive review delves into the molecular mechanisms underlying the impact of abiotic stresses on rice growth and development, and explores the use of various omics analyses, such as transcriptomics, proteomics, and metabolomics, to identify potential candidate genes linked to abiotic stress. The review is well-organized and provides detailed insights.
However, to further enhance the review's impact and utility, I recommend that the authors include an extensive table featuring all the classical/cloned genes and their genome positions in both the older and newer reference genomes, along with corresponding citations. This would not only attract more citations, but also enable researchers to locate all the genes mentioned in the review and explore their relevance to new genomic regions they may be investigating for a particular trait.
There are a couple of grammatical errors.
Author Response
The authors' comprehensive review delves into the molecular mechanisms underlying the impact of abiotic stresses on rice growth and development, and explores the use of various omics analyses, such as transcriptomics, proteomics, and metabolomics, to identify potential candidate genes linked to abiotic stress. The review is well-organized and provides detailed insights. Discuss the implications of the findings for the field and highlight their potential impact.
We would like to thank the reviewer for positive and insightful comments on the manuscript. We believe this input has been invaluable to make our manuscript more balanced. We have taken the comments on board to improve and clarify the manuscript. Below is our response to the comments raised in the review.
Comment: However, to further enhance the review's impact and utility, I recommend that the authors include an extensive table featuring all the classical/cloned genes and their genome positions in both the older and newer reference genomes, along with corresponding citations. This would not only attract more citations, but also enable researchers to locate all the genes mentioned in the review and explore their relevance to new genomic regions they may be investigating for a particular trait.
Response: Thanks for the nice suggestion. We have inserted a table featuring key genes mentioned in this review. We added their Locus ID and genome positions following two different databases.
We’ve checked the whole manuscript again and removed the possible typo, sentence, grammatical, and other possible mistakes. We hope that this version of the manuscript is satisfactory and readers friendly. We again thank the reviewer for the nice comments.
Round 2
Reviewer 1 Report
No further comments. Manuscript reads better now.
Minor editing